# High-resolution spatial prediction of anemia risk among children aged 6 to 59 months in low- and middle-income countries

## Abstract

**Background** Anemia, a severe condition among children associated with adverse health effects such as impaired growth, limited physical and cognitive development, and increased mortality risk, remains widespread, particularly in low- and middle-income countries. This study combines Demographic and Health Surveys data with remotely sensed climate, demographic, environmental, and geo-spatial information, creating a data set comprising about 750,000 observations on childhood anemia from 37 countries. It is used to provide high-resolution spatio-temporal estimates of all forms of childhood anemia between 2005 and 2020.

**Methods** Employing full probabilistic Bayesian distributional regression models, the research accurately predicts age-specific and spatially varying anemia risks. These models enable the assessment of the complete distribution of hemoglobin levels. Additionally, this analysis also provides predictions at a high resolution, allowing precise monitoring of this indicator, aligned with Sustainable Development Goal (SDG) 2.

**Results** This analysis provides high-resolution estimates for all forms of anemia and reveals and identifies striking disparities within and between countries. Based on these estimates, the prevalence of anemia decreased from 65.0% [62.6%–67.4%] in sub-Saharan Africa and 63.1% [60.6%–65.5%] in South Asia in 2010 to 63.4% [60.7%–66.0%] in sub-Saharan Africa and 58.8% [56.4%–61.3%] in South Asia in 2020. This translates into approximately 98.7 [94.5–102.8] million and 95.1 [91.1–99.0] million affected children aged 6 to 59 months in 2020, respectively, making it a major public health concern.

**Conclusions** Our approach facilitates the monitoring of age-specific spatio-temporal dynamics and the identification of hotspots related to this important global public health issue. To our knowledge, this represents the first high-resolution mapping of anemia risk in children. In addition, these results reveal striking disparities between and within countries and highlight the influence of socio-economic and environmental factors on this condition. The findings can guide efforts to improve health systems, promote education, and implement interventions that break the cycle of poverty and anemia.

## Plain language summary

Anemia, a condition where the body lacks enough red blood cells to carry oxygen, causes symptoms like tiredness and weakness. It remains widespread, especially in low- and middle-income countries, particularly in sub-Saharan Africa and South Asia. Researchers use advanced computational methods, including machine learning, to assess anemia severity and predict outcomes across regions. These methods reveal significant disparities in anemia prevalence within and between countries, influenced by socio-economic and environmental factors. The findings can guide targeted health interventions, improving health systems, and breaking the cycle of anemia and poverty. By tackling anemia, we can make progress toward global goals, such as those set in the 2030 Agenda for Sustainable Development.

Anemia is a condition that impairs the blood's ability to transport oxygen effectively. It is characterized by a decreased number of erythrocytes (red blood cells) or a hemoglobin (Hb) level below a specific cutoff, which varies based on factors such as age, sex, smoking behavior, and elevation[1–4]. Anemia remains a serious public health concern that affects about 40% of children below five years of age and about one-third of all women of reproductive age worldwide[4]. Anemia is particularly prevalent in low- and middle-income countries (LMICs), with the highest burden observed in South Asia and sub-Saharan Africa[2,5,6]. In these regions, anemia frequently coexists with a high prevalence of stunting, which refers to a low height-for-age ratio[7].

✉e-mail: johannes.seiler@uibk.ac.at

Anemia has substantial negative health consequences, particularly among vulnerable populations such as young children and women, who are particularly susceptible during menstruation and pregnancy[2,4]. Besides its direct negative health effects, anemia can impede physical and cognitive development, leading to detrimental consequences for educational achievements and economic productivity[1,4].

Anemia is multifactorial and can be caused by nutrient deficiencies through inadequate diets, for example, a lack of vitamin A, a lack of iron or vitamin B12, inadequate absorption of nutrients, or infections (e.g., malaria, parasitic worm infections, tuberculosis, and HIV), with iron deficiency being one of the most common reasons[2,8,9]. Hence, anemia is often closely linked to malnutrition and food insecurity and can be considered an indicator of the prevalence of "hidden hunger", which refers to the chronic deficiency of essential vitamins and minerals. Hence, being indicative of both poor overall health and inadequate nutrition—in particular for hidden hunger—anemia is also included as a child-specific indicator outlined in Sustainable Development Goal (SDG) 2.2—*Malnutrition: End all forms of malnutrition*—which focuses on achieving zero hunger, including addressing hidden hunger[4]. However, anemia has received comparatively less attention than anthropometric assessments of malnutrition.

In recent years, several studies have been conducted that combine survey data with geo-spatial information to map various health outcomes and to analyze their spatio-temporal patterns. These studies have explored child health outcomes like chronic and acute malnutrition or child mortality rates[10–16], maternal health [17], and infectious diseases[18–23]. However, when it comes to childhood anemia, studies that provide high-resolution estimates of the burden of anemia and that analyze potential determinants of anemia are limited to single countries[24–27] or small groups of countries[28,29]. Estimates of levels and trends across geopolitical regions are limited to the study by Stevens and colleagues[30], which provides an almost global assessment of anemia risk at the country level and is shared by the World Health Organization (WHO,[31]). Thus, the knowledge of the levels and trends beyond country estimates of anemia in LMICs is still limited to individual countries, or small groups of countries. This highlights the need to enhance our understanding of the prevalence of anemia, its underlying causes, and its associations with other health outcomes in LMICs. This information is essential for designing targeted interventions and allocating scarce resources most cost-effectively to address this global health problem.

This study contributes to the existing literature by extending the analysis from a single-country to a multi-country-year analysis allowing to estimate spatio-temporal dynamics of anemia. By employing Bayesian distribution regression models, the study enables the characterization of the complete distribution of Hb levels of children below the age of five and facilitates the creation of comprehensive spatio-temporal maps of all forms of anemia risk. Since different regions and populations may have distinct risk factors and determinants of anemia, studying the prevalence of anemia in sub-Saharan Africa and South Asia at a detailed level allows for the identification of high-risk areas and specific population subgroups. In addition, this framework allows to determine the association of anemia with climatic, environmental, and socio-economic factors using a unique data set for 37 LMICs. In doing so, we will provide valuable insights into the spatio-temporal distribution of this important global public health determinant and its association with climatic, environmental, and socio-economic correlates.

Striking disparities in the prevalence of anemia are revealed both between and within countries, with socio-economic and environmental factors playing a key role. In particular, regions such as the Sahel zone in West Africa (e.g., Burkina Faso and Mali) and parts of Ethiopia, Kenya, and Mozambique in East Africa emerge as significant hotspots, exhibiting higher-than-average prevalence rates. The analysis also highlights substantial non-linearities for certain factors, suggesting that the effects of these covariates on anemia may vary depending on their levels. The findings can guide efforts to improve health systems and implement interventions to break the cycle of poverty and anemia.

## Methods

The analytical framework for estimating the prevalence of any form of anemia can be summarized as follows and is described in more detail in the Supplementary Methods. The five distinct steps are briefly outlined below: preprocessing the data, identifying appropriate continuous response distributions, identifying informative covariates, re-estimating the final model, and model validation and calibration diagnostics.

### Data

Data from several different sources were merged into one large data set to be used in the analysis. The data sources and preprocessing steps are described below and in Supplementary Table 1, and Supplementary Fig. 1.

**Demographic and Health Surveys**. Measurements of the Hb level are extracted from the children's recode of the Demographic and Health Surveys (DHS) data, which provides the core data used in this study. The DHS data collection[32] has been conducted by ICF International in over 90 LMICs[33]. The cross-sectional surveys use a multistage stratified sampling design and are representative at least at the first administrative level. They collect data regarding the health and well-being of women of reproductive age (typically aged 15–49 years), their children (born within the past five years of data collection), their partner, and their household. Most surveys after 2000 include geo-coded information on the location of the primary sampling unit (PSU) a household pertains to. A key advantage of the DHS is the availability of comparable data for multiple countries and consistent quality of reporting and data over time. Additional details on the DHS surveys are available, e.g., from Corsi and colleagues[34].

**Additional data sources**. To enable spatial extrapolation to unsurveyed locations, as well as to account for socio-economic and demographic covariates in the DHS data, this study incorporates remotely sensed information. In doing so, the individual-level survey data are merged with geo-spatial and remotely sensed data from various other sources such as the Malaria Atlas Project (MAP), or the Uppsala Conflict Data Program (UCDP). Supplementary Table 1 provides information on the data sources. The factors considered include those that have a direct association with anemia (such as altitude or malaria incidence) as well as those that may potentially be linked to anemia (such as temperature, land cover, distance to bodies of water, or night-time light). The incorporation of these factors allows for the prediction of anemia prevalence in locations where direct estimates are not available. Detailed information on the covariates included is presented in Supplementary Table 3.

### Measurement of anemia

The Hb level, which is measured by taking a few droplets of blood, is commonly used as a biomarker to determine the anemia status of the sampled individuals[35]. This biomarker is routinely collected by the DHS[32], which is used as the main data source. Commonly, the DHS relies on the HemoCue system for Hb measurement, which uses a capillary blood sample from the children to measure the Hb level[35].

Anemia is a condition in which the Hb level in the blood is below a predefined age-, elevation-, and sex-specific threshold[1,2]. After adjusting for altitude, the WHO[3,9] defines the age-specific Hb threshold for anemia in children between 6 and 59 months to be less than 110 g$L^{-1}$. In addition, within this subpopulation, Hb levels between 100 and 109 g$L^{-1}$ are considered mild, levels between 70 and 99 g$L^{-1}$ are considered moderate, and levels below 70 g$L^{-1}$ are considered severe[1]. Other sources rely on slightly different thresholds[1]. See Supplementary Note 2 and the Supplementary Methods for more information on the input data and how the input data was split into training and test data. Detailed information on the data sources, summary statistics, and the individual preprocessing steps can be found in the Supplementary Methods, Supplementary Tables 1 and 2, and Supplementary Fig. 1. Furthermore, see Supplementary Note 2 for a discussion on

**Fig. 1 | Scatter plot of the survey-based prevalence of anemia reported by DHS and the model-based anemia prevalence.** The reported anemia prevalence and the model-based anemia prevalence are aggregated to admin-2 regions within countries and for children aged 6–59 months for: **a** any anemia (i.e., P(Hb < 110 gL$^{-1}$); **b** mild anemia (i.e., P(100 gL$^{-1}$ ≤ Hb < 110 gL$^{-1}$)); **c** moderate anemia (i.e., P(70 gL$^{-1}$ ≤ Hb < 100 gL$^{-1}$)); and **d** severe anemia (i.e., P(Hb < 70 gL$^{-1}$)). In all panels, in addition, the correlation coefficient $\rho$ is reported for the different sub-regions.

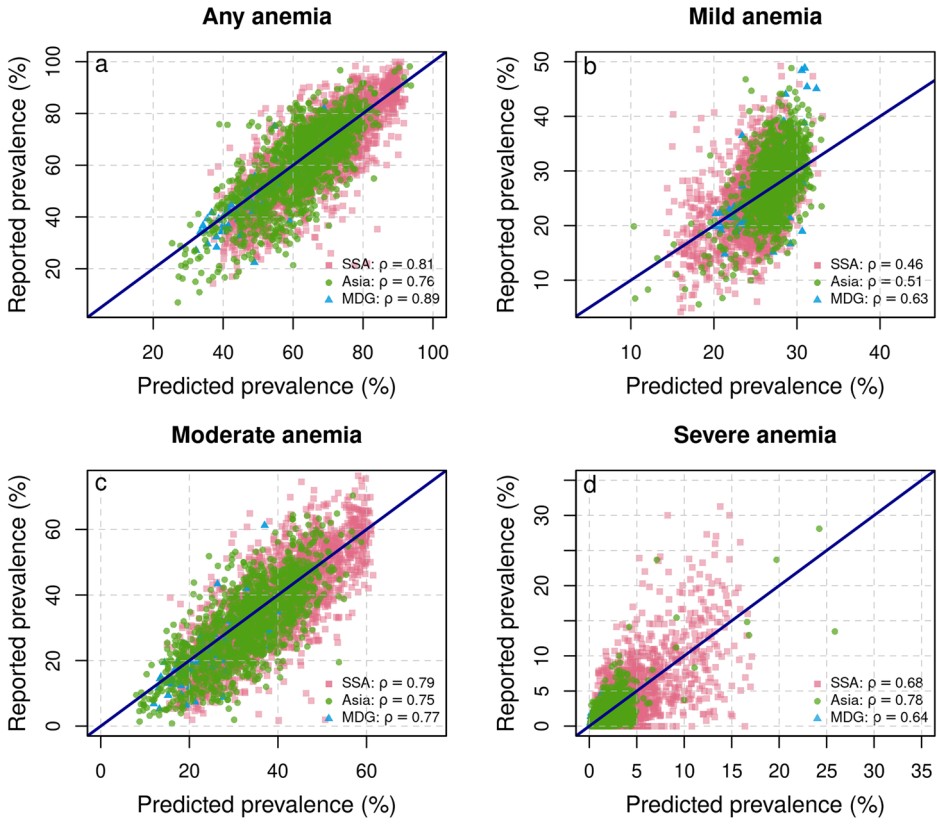

influencing factors of anemia. In addition, Supplementary Note 1 gives a brief overview of the different Supplementary Items provided in addition to the main manuscript.

### Distribution of Hb levels

The choice of the distributional model is crucial when developing a complete probabilistic distributional regression model that can relate Hb levels to covariates. One commonly employed method for estimating Hb levels is based on the assumption that these levels within the population follow a normal distribution (NO) and can be effectively modeled as such[28,29]. Besides binary regression models, models assuming a normal distribution are common in many applications[27,28]. Nevertheless, reliance on these models can introduce estimation errors when the chosen distribution poorly aligns with the characteristics of the data[36] (see also Supplementary Fig. 4).

### Probabilistic distributional regression

To accurately estimate all risk levels of anemia, it is necessary to use a full probabilistic model that explains the complete distribution of Hb levels based on relevant covariates. To characterize the complete distribution of the Hb level and select important correlates determining the Hb level, we employ Bayesian estimation techniques based on Markov chain Monte Carlo (MCMC) simulations[37]. Incorporating geo-coded information allows analyzing the spatio-temporal dynamics of anemia risk while maintaining the interpretability of the estimated effects[22]. The two main regions (i.e., South Asia, and mainland sub-Saharan Africa) are analyzed separately due to geographic disconnectedness. Further note that Madagascar is treated as its own region, as, for example, Lai et al.[22] points out that the interpretation of spatial correlation between sampled observations differs when separated by water compared to spatial correlation over dry land. See Supplementary Fig. 2 for a map of the analyzed regions and the locations of the primary sampling unit (PSU). For a detailed description of the modeling approach, see Supplementary Methods.

### Variable selection

Using a highly efficient boosting algorithm, which is described in detail in ref. 38, relevant covariate terms are selected from the full set of covariates included in the model, as outlined in Supplementary Table 3. Supplementary Fig. 16 shows the selected terms of each distributional parameter and the corresponding selection frequencies separately for each region. Among the covariates selected in each region are the wealth index, the age in months, the elevation at the PSU, the air temperature (measured 2 m above the surface), and the location of the PSU. It is particularly worth mentioning that previous studies[28,29,39,40] have also emphasized the importance of these covariates. More decisive, although a complex location-scale model is estimated, the spatial components are still frequently selected, which highlights that heterogeneity is not yet fully explained by the included covariates.

### Calibration and convergence checks

To assess the predictive performance, the data set is split into a training data set containing 80% of the data, and a test data set of 20%, which is used for model validation only—and in particular to assess the predictive performance—applying the methods described in the Supplementary Methods. Further details on the model diagnostic are provided in Supplementary Note 3.

The final models are well-calibrated as shown by the corresponding probability integral transform (PIT) histograms (Supplementary Fig. 5 to Supplementary Fig. 7). Additional regression diagnostics, such as convergence checks are provided in Supplementary Fig. 8 to Supplementary Fig. 11. Furthermore, as depicted in Fig. 1, the predictions derived from the model show a strong correlation with the descriptive regional admin-2 level estimates obtained directly from DHS. As illustrated by this figure the model-based predictions—aggregated to the admin-2 level—show for all different types of anemia a very strong linear correlation (up to $\rho = 0.89$) with the descriptive admin-2 level prevalence directly calculated using the DHS as input data. Taking all this into consideration underscores the reliability of the model-generated estimates to produce reliable

estimates. See also Supplementary Fig. 12 to Supplementary Fig. 15 for further in- sample, as well as out-of-sample validation checks of the categorical anemia prevalence estimates and the continuous Hb levels. An additional check for robustness and validation of the model's predictive performance and generalizability is provided by examining the 95% posterior prediction interval (PPI) computed on out-of-sample data. If the model has good predictive properties, approximately 95% of the observed out-of-sample data should lie within the 95% PPI. Supplementary Table 4 confirms this, showing that for each subregion, approximately 95% of observed out-of-sample data points fall within the 95% PPI bounds, underscoring the model's strong predictive performance.

### Ethical approval

This study only involved secondary analysis of publicly available DHS data sets, which were de-identified and merged with non-human data. As such, it did not require additional research ethics approval. The DHS data sets are accessible after registration from the DHS data collection[32] and comply with ethical guidelines for research involving human subjects, as outlined in the *Protecting the Privacy of DHS Survey Respondents* documentation[41]. The surveys have been reviewed and approved by ICF's Institutional Review Board (IRB), and country-specific protocols are also reviewed by both the ICF International IRB and an IRB in the host country. An informed consent statement is read to each respondent, who may accept or decline. For participating children, a parent or legal guardian must provide informed consent prior to participation. Therefore, no further IRB approval is necessary for studies using DHS data. All other data sources used in this study do not include human subject's data.

### Reporting summary

Further information on research design is available in the Nature Portfolio Reporting Summary linked to this article.

## Results

### Effect of covariates (marginal probabilities)

Figure 2 illustrates the marginal prevalence along with 95% credible intervals of severe, moderate, and mild anemia for selected covariates. The first row presents the marginal prevalence of these anemia categories based on the age of the child. The second row displays the marginal prevalence related to household wealth, while the last row depicts the association between the surface temperature and the marginal anemia prevalence. The locations are chosen to illustrate that the prevalence can be estimated based on the model for any location.

The analysis of the age-specific effect on the prevalence of anemia reveals three noteworthy aspects. First, there is a highly non-linear relationship between child age and anemia prevalence. This highlights the importance of accounting for non-linearities in the model, as neglecting them would lead to a mis-specified model and inaccurate predictions. Second, the magnitude of the age effect is also substantial relative to the other covariates included. This underscores the importance of stratifying estimates by age of the child to produce age-specific spatio-temporal risk maps of anemia prevalence. Third, the prevalence of anemia appears to be highest during the first two years of a child's life—a critical period in their development—then the change in prevalence of any form of anemia steadily decreases and appears to be less pronounced, as indicated by the flattening gradient. Note that the DHS does not provide information on the Hb levels in children under six months of age. This is because Hb levels are usually higher at birth and within a few months after birth, possibly due to better provision of nutrients in-utero, which may bias the estimated prevalence of anemia[35]. With increasing age, the prevalence continues to decline, although the rate of decline varies depending on the location.

Anemia and household wealth are negatively correlated, although the effect is less pronounced than the age effect, with the prevalence being highest among children living in poor households. The correlation between anemia and household wealth suggests that food scarcity and lack of

nutritious staple foods may contribute substantially to adverse health outcomes among members of disadvantaged households. This association underscores the importance of addressing socio-economic factors and improving access to nutritious food to reduce the negative impact of anemia and other health-related outcomes.

When considering remotely sensed environmental and climate-specific covariates, we find that depending on the region different covariates have been selected, while elevation and temperature consistently appear across all regions (Supplementary Fig. 16). Examining the bottom panel of Fig. 2, which shows the estimated effect of surface temperature, along with 95% credible intervals on the prevalence of any form of anemia for selected locations. A relatively constant effect is observed in mainland sub-Saharan Africa and Madagascar. In contrast, in South Asia, there is a shallow but steadily increasing association with increasing surface temperature, suggesting that within South Asia, the prevalence of any form of anemia increases with increasing surface temperature. Furthermore, the estimated credible intervals within South Asia are narrower than the credible intervals in sub-Saharan Africa, highlighting the importance of this effect within South Asia. Accordingly, in areas expected to be severely affected by climate change, this relationship and potential pathways by which rising temperatures might lead to an increase in anemia prevalence warrants close monitoring to better understand the impact of rising temperatures and their mechanisms on anemia prevalence.

### Temporal trends at different administrative levels

For the period from 2010 to 2020, based on the model-based estimates 29 (i.e., 58%) of the 50 countries for which we obtained projections of anemia prevalence showed a decrease in the prevalence of anemia of more than 2%, in 10 countries (i.e., 20%) the prevalence of anemia remained more or less constant ranging between −2% and 2%, and in 11 countries (i.e., 22%) the model-based estimates predicted an increase in the prevalence of anemia of more than 2%. This observation is also summarized in Supplementary Fig. 3 where the country-level time trends are plotted using the survey data. See also Supplementary Table 5 to Supplementary Table 8 showing the nationally-aggregated projections of any anemia, mild anemia, moderate anemia, and severe anemia based on the pixel-level model-based estimates for the years 2010, 2019, and 2020, together with the corresponding WHO estimates of anemia[30,31] and the estimated population at risk taken from the United Nations World Population Prospects[42].

At smaller administrative units a similar pattern emerges. At the admin-1 level from the 713 included admin-1 level regions for which model-based estimates have been obtained about 55% of the regions showed a decrease in the prevalence of anemia, in about 27% of regions the prevalence of anemia remained more or less constant, and in about 18% of the regions the prevalence of anemia increased. Moreover, as shown in Fig. 3, across countries and within countries, the inequality varies greatly, with a tendency for countries with a high prevalence of anemia to have a lower within-country variability.

### Spatio-temporal dynamics

Figure 4 illustrates the overall anemia prevalence disaggregated at the pixel level (20 × 20 km), highlighting distinct spatio-temporal patterns and hotspots, in particularly in the western part of sub-Saharan Africa in countries like Burkina Faso, where the model-based estimates project that the prevalence of anemia remained high, and decreased from 87.3% [95% credible interval 86.4%–88.2%] in 2010 to 77.1% [75.7%–78.4%] in 2020 and similar in Mali, where the model-based estimates project that the prevalence of anemia decreased from 84.3% [83.1%–85.5%] in 2010 to 77.4% [75.7%–79.0%] in 2020. Also in the eastern part of sub-Saharan Africa in countries such as Mozambique high levels of anemia are observed, e.g., the model-based estimates project that the prevalence of anemia in Mozambique remained more or less constant at 71.9% [70.1%–73.6%] in 2010 compared to 72.7% [70.6%–74.7%] in 2020 or Kenya, where the prevalence of anemia remained more or less constant at 62.6% [60.7%–64.5%] in 2010 to 62.5% [60.4%–64.6%] in 2020 a similar pattern emerges. Figures 5 and 6

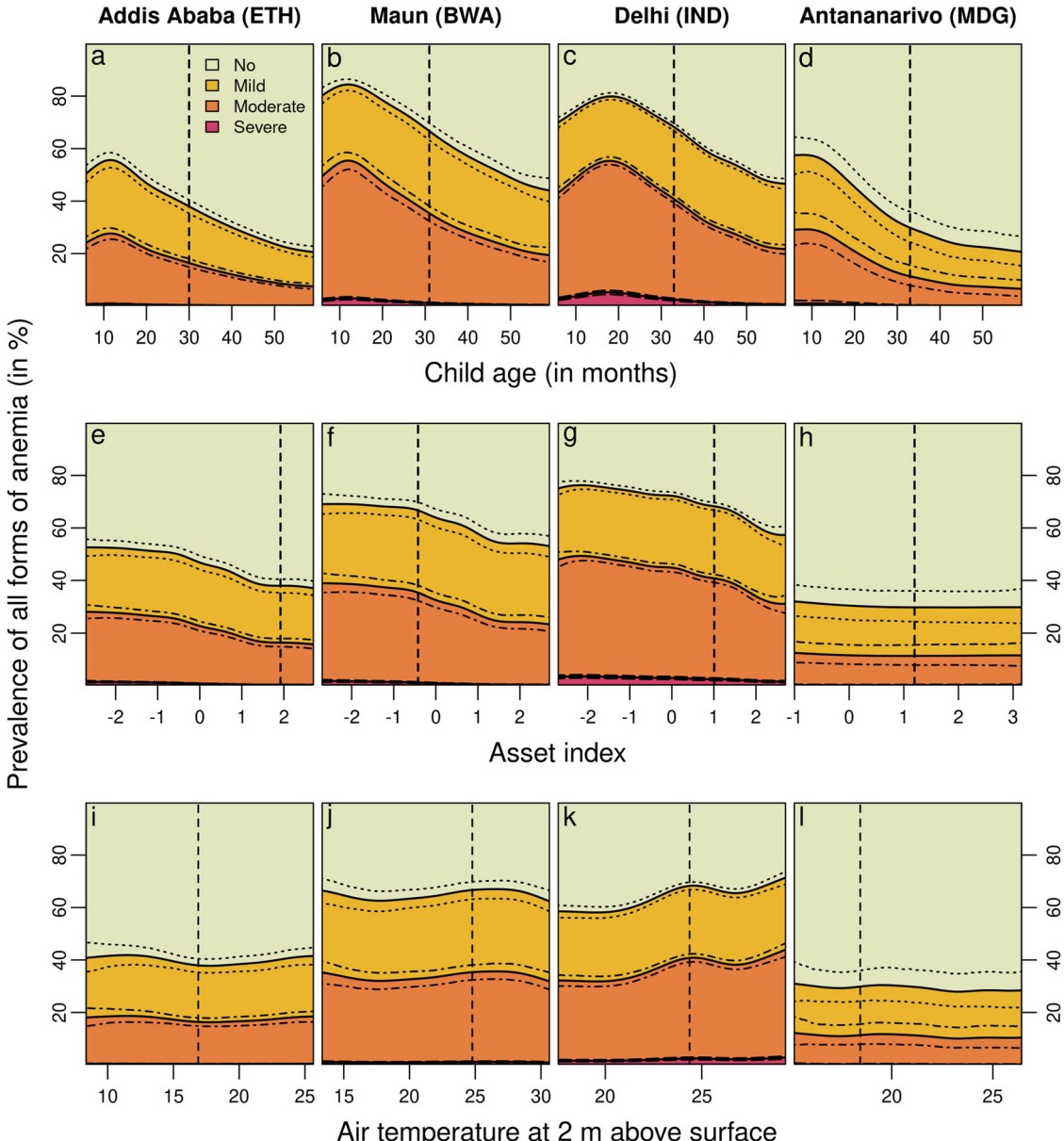

**Fig. 2 | Marginal prevalence of severe, moderate, and mild anemia among children aged 6–59 months for selected covariates and locations. a–d** illustrate the marginal prevalence of any form of anemia for the age of the children at the four locations Addis Ababa, Maun, Delhi, and Antananarivo; **e–h** show the effect of the household's wealth index on the prevalence of any form of anemia at the four aforementioned locations; **i–l** depict the estimated effect of the temperature on the marginal prevalence of any form of anemia at the four aforementioned locations. To estimate the marginal prevalence at a given location, values for numeric covariates are fixed at the median observed within a buffer of 20 km around the location, and values for categorical covariates are fixed at the mode observed within a buffer of 20 km around the given location, respectively. Vertical dashed lines indicate the median observed for the illustrated covariate at the given location. The dotted curves around the solid lines represent the 95% credible interval of the corresponding anemia prevalence estimate. ISO3 country codes are used as identifiers, where ETH stands for Ethiopia, BWA for Botswana, IND for India, and MDG for Madagascar.

illustrate the estimated prevalence of overall, mild, moderate, and severe anemia for different age bins without stratification by sex over time. The corresponding risk maps depicting the spatio-temporal dynamics stratified by sex are shown in Supplementary Fig. 18 to Supplementary Fig. 23, respectively.

In contrast to the declining prevalence of chronic and acute malnutrition indicators such as stunting, underweight, and wasting over the past 20 years[15,16], the prevalence of anemia has remained constant or showed only modest improvement. This observation is also supported by Supplementary Fig. 3, which shows the time trend of the prevalence of anemia at the country level. Especially after 2010, the model-based estimates project that about 22% of the included countries showed an increase in the

prevalence level of anemia that exceeds 2%. This suggests that anemia may be the result of a complex and multifactorial disease history[17,30]. It also indicates that there may be a correlation between populations with a high prevalence of anemia and those with a high prevalence of the diseases that cause anemia. Furthermore, it highlights that this population group may face certain disadvantages.

The analysis provides evidence of the substantial burden of anemia in children aged 6–59 months, which is estimated to affect about 98.7 [94.5–102.8] million children in sub-Saharan Africa and about 95.1 [91.1–99.0] million children in South Asia in 2020. In particular, the eastern and western regions of sub-Saharan Africa and parts of South Asia the model-based estimates project high anemia prevalence for the year 2020,

**Fig. 3 | Within-country geographical inequality in anemia (i.e., P(Hb < 110 gL$^{-1}$)).** Inequality estimates are shown across 50 countries among children aged 6–59 months for: **a** the year 2010, and **b** the year 2020. The panels show the absolute within-country inequalities at the admin-2 level across 50 countries. Each dot represents an admin-2 region. The lowest (highest) dot of each bar corresponds to the admin-2 level region with the lowest (highest) estimated anemia prevalence in each country. The horizontal line in each bar is the estimated anemia prevalence at the country level. Note that ISO-3 country codes are used as abbreviations.

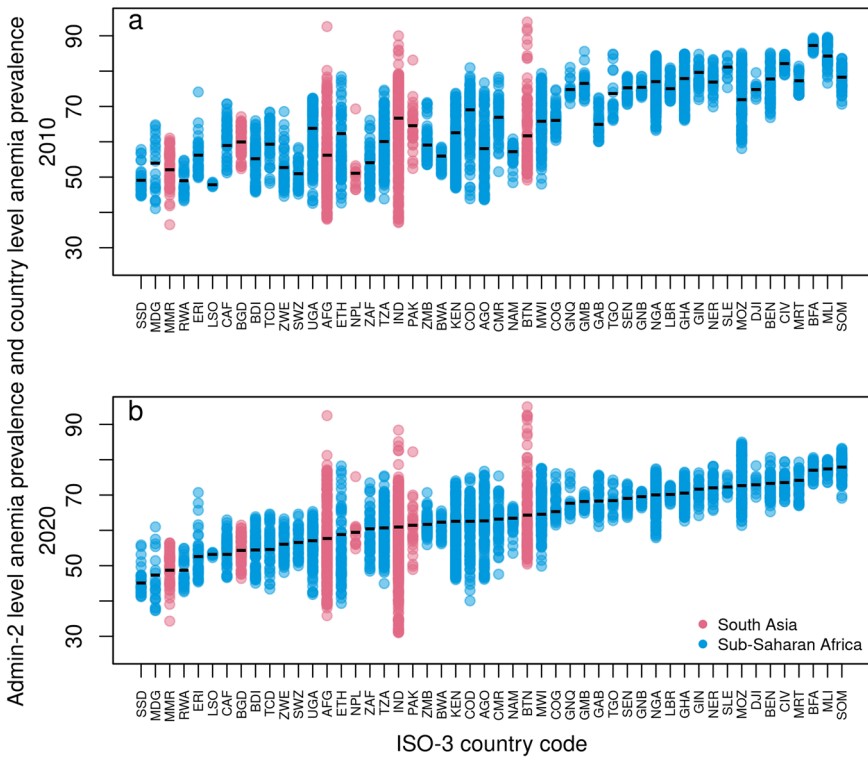

with varying hotspots (e.g., in sub-Sahara Africa in the admin-1 regions Sud-Ouest 79.8% [78.6%–81.1%] (Burkina Faso); Boucle du Mouhoun 79.6% [78.3%–80.8%] (Burkina Faso); Vallée du Bandama 79.4% [77.9%–80.8%] (Côte d'Ivoire); Ségou 78.7% [77.1%–80.2%] (Mali); Zambezia 79.8% [78%–81.5%] (Mozambique); Nampula 79.6% [77.4%–81.6%] (Mozambique); in South Asia in the admin-1 regions Bamyan 74.8% [69.8%–79.4%] (Afghanistan); Gujarat 69.7% [67.7%–71.8%] (India); Northern Areas 82.2% [79.3%–85.0%] (Pakistan)).

Contrary and gratifying, in sub-Saharan Africa and South Asia the model-based estimates project also coldspots with a low prevalence of anemia in the year 2020 (e.g., in sub-Sahara Africa in the admin-1 regions. Antananarivo 38.5% [32.3%–45.2%] (Madagascar); Toamasina 41.2% [34.4%–48.4%] (Madagascar); Kisoro 41.8% [40%–43.7%] (Uganda); Kabale 41.2% 43.1% [41.3%–44.8%] (Uganda); in South Asia in the admin-1 regions Manipur 32% [30%–33.9%] (India); Nagaland 36.7% [34.7%–38.7%] (India); Chin 36.9% [34.5%–39.2%] (Myanmar)). Together this indicates that high disparities between and within countries exist.

Furthermore, Fig. 5 and Supplementary Figs. 20 and 21 highlight that the prevalence is highest among young children at approximately six months of age. The prevalence gradually decreases non-linearly with increasing age and is the lowest for children of 59 months of age. This also aligns with the patterns depicted in the first row of Fig. 2.

Similar to other indicators of chronic and acute malnutrition in the year 2020, countries with an exceedingly high anemia prevalence exist. These hotspots are predominantly located in western sub-Saharan African countries, including Burkina Faso (projected prevalence of anemia in 2020 77.1% [75.7%–78.4%]), Mali (projected prevalence of anemia in 2020 77.4% [75.7%–79.0%]), Niger (projected prevalence of anemia in 2020 72.0% [70.0%–74.0%]), and Nigeria (projected prevalence of anemia in 2020 70.1% [68.2%–71.9%]). Additionally, hotspots are observed in eastern sub-Saharan Africa, encompassing countries such as Kenya (projected prevalence of anemia in 62.5% [60.4%–64.6%]), Somalia (projected prevalence of anemia in 77.9% [75.3%–80.3%]), and Mozambique (projected prevalence of anemia in 2020 72.7% [70.6%–74.7%]). These findings align with existing literature that highlights the specific concern regarding low levels of Hb in western parts of sub-Saharan Africa[9]. It is noteworthy that

Madagascar (projected prevalence of anemia in 2020 47.3% [40.2%–54.8%]), despite having one of the highest prevalence rates of chronic and acute malnutrition[15,43], exhibits comparatively lower prevalence rates of anemia compared to other countries within sub-Saharan Africa.

To visualize both the spatial and temporal dynamics within different countries, Supplementary Fig. 17 depicts the trends over time in Nigeria, Ethiopia, and Bangladesh. In Nigeria, childhood anemia improved; however, it remains at a high level, particularly noticeable in some coastal regions and the northwest of the country. While some regions in Ethiopia have successfully reduced childhood anemia levels between 2010 and 2015, anemia stagnated between 2015 and 2020 at a high level. For example, Bangladesh's childhood anemia remained constant from 2010 to 2015, followed by a slight decrease in 2020. However, the overall prevalence of moderate childhood anemia in Bangladesh remains high.

## Discussion

Before discussing our main findings and placing them in a broader context, we also want to address and mention some limitations of our study. First, the measured Hb levels may have been measured or reported erroneously. This may be due to malfunction, misuse of measurement equipment, or transmission errors. Even though the DHS are known for their high standards of data quality, this point cannot be ruled out. Second, hypothetically, an individual eligible for Hb measurement who normally resides above 1000 m above sea level was below 1000 m above sea level in the weeks before the measurement (or vice versa), which could potentially distort the measured Hb level. Third, the uncertainty (see e.g., Figs. 2 and 4) is particularly high in areas where data are scarce, in areas that are not densely populated (e.g., desert-like areas in South Africa and Namibia), or where the sample size is small (e.g., Madagascar); this usually also underscores the point that newer and updated survey statistics in these regions are urgently needed to closely monitor the progress toward the SDGs in these areas. Fourth, although we have taken great care in the pre-processing steps to merge the data sources, some of the input data are pre-existing (highly modeled) estimates (e.g., malaria prevalence), which are themselves subject to inaccuracies. To our knowledge, there are yet no methods to account for measurement error

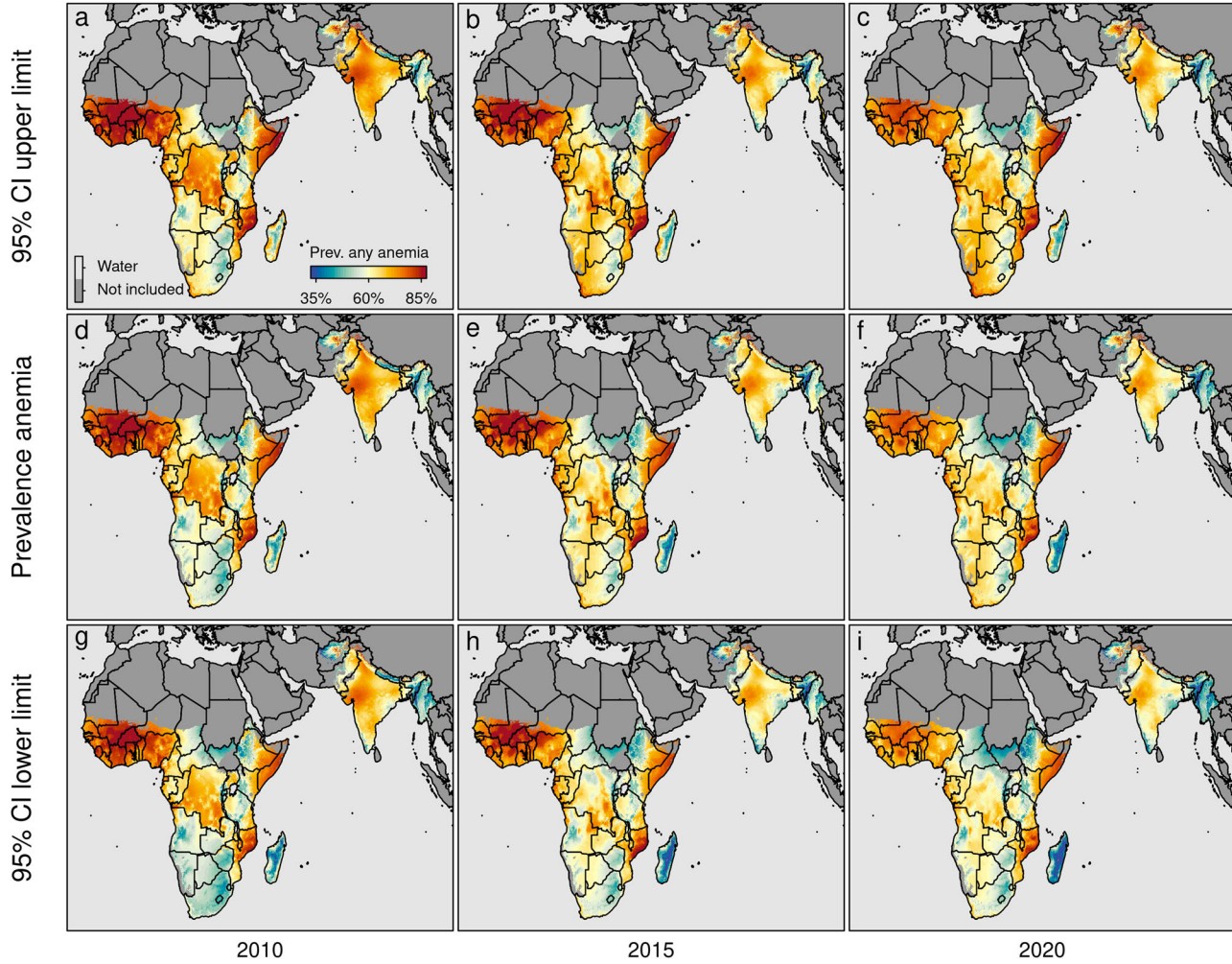

**Fig. 4 | Predicted marginal spatio-temporal prevalence of any form of anemia (i.e., P(Hb < 110 gL⁻¹)).** The maps show the estimated anemia prevalence among children between 6 and 59 months (**d–f**) together with 95% credible intervals (**a–c, g–i**) for the years 2010, 2015, and 2020. Boundaries reflect administrative boundaries at the country level. Pixels categorized as *Barren*, or *Permanent Snow and Ice*, and pixels above 3750 m (1900 m) of altitude in sub-Saharan Africa and South Asia (Madagascar) are flagged as *Not included*.

in GAMLSS models, so the estimated uncertainty may underestimate the true uncertainty.

Merging geo-referenced survey data with remotely sensed data from several distinct sources allows for the first time to assess the spatio-temporal risk of all forms of anemia among children in sub-Saharan Africa and South Asia—the two regions affected disproportionally by this major public health issue—using fully probabilistic distributional regression. Rigorously assessing the predictive performance of this method allows us to predict the Hb level at locations where no direct measurements are available. The excellent predictive skills of this approach are highlighted in Fig. 1, where the regional predicted prevalence at admin-2 level are plotted against the reported prevalence at the admin-2 level. The correlation—depending on the region and the type of anemia—is in most cases greater than 0.75 highlighting the excellent prediction capability of this method at the sub-national level. Moreover, the model-based estimates aggregated to the country level show a very high degree of overlap with the country level estimates of Stevens et al.[30] that are provided by the WHO[31] until the year 2019. See also Supplementary Table 5 to Supplementary Table 8 for details on the model-based estimates aggregated to the country level and the WHO country level estimates. However, there are some differences between the model-based estimates and the WHO estimates, for example, the WHO estimates are more optimistic for India, Nepal, Pakistan, and Somalia. However, the model-based estimates include more recent DHS surveys in which the prevalence of anemia has increased or plateaued compared with previous surveys[44,45]. In contrast, the Stevens et al.[30] and WHO estimates are more conservative for countries such as Chad, the Central African Republic, or South Sudan. These are also countries for which recent DHS data with Hb measurements are not available, and thus our model predicts anemia prevalence based on the included covariates and the estimated spatio-temporal association. This implies that the model-based associations between the included covariates and Hb levels, and the observed spatio-temporal dynamics are similar between included countries and countries without direct ground observations. To investigate whether these estimates are nevertheless plausible, a comprehensive literature review aims to examine them more closely for plausibility. For a comprehensive comparison of the estimates, see Supplementary Table 9. Overall, the estimates of anemia prevalence at the country level from this study fall within the uncertainty intervals or overlap substantially in most cases. In some cases, however, there are discrepancies between some country-level studies and surveys, the Stevens et al.[30] and WHO estimates, and the model-based estimates presented in this research article. However, this highlights two important aspects that should be emphasized: First, all of the included case reports at the country level that are based on studies or surveys indicate an anemia prevalence well above 40%—i.e., the WHO cut-off for indicating a serious public health problem in this context. This means that anemia is certainly a serious public health problem in these countries. Second, the discrepancies highlight the importance of the

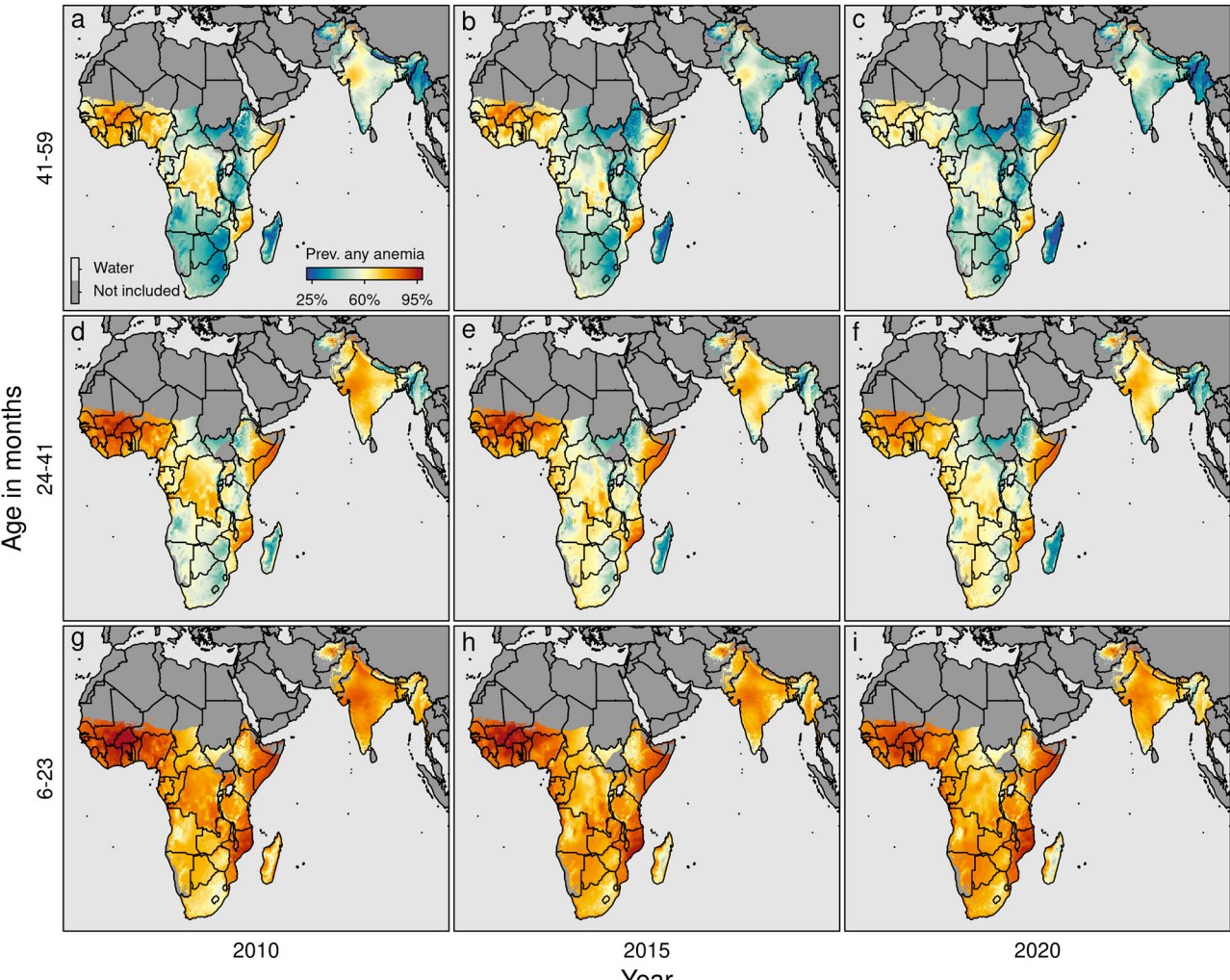

**Fig. 5 | Predicted marginal spatio-temporal prevalence of any form of anemia (i.e., P(Hb < 110 gL⁻¹)).** The maps show the estimated anemia prevalence among children aged 6–59 months for the years 2010, 2015, and 2020 for the following three age bins: (**a**–**c**) (42, 60] months; (**d**–**f**) (24, 42] months; and (**g**–**i**) (6, 24] months, respectively. The *x*-axis shows the dynamics over time, the y-axis shows the age-specific dynamics of the overall prevalence of anemia. Note that the non-geo-referenced covariates are fixed at the median (metric covariates) or the mode (categorical covariates), respectively. Boundaries are at the country level. Pixels categorized as *Barren*, or *Permanent Snow and Ice*, and pixels above 3750 m (1900 m) of altitude in sub-Saharan Africa and South Asia (Madagascar) are flagged as *Not included*.

need for better and more accurate surveillance, including more frequent geo-referenced surveys. This would allow better monitoring, and targeting of the population at risk.

Our results are consistent with the findings of other studies on the risk factors of anemia[29,40], and with studies on risk factors of other forms of health outcomes related to nutrition[13,16,46]. Another intriguing aspect is the inclusion of malaria incidence as a selected covariate solely in sub-Saharan Africa and Madagascar. This choice reflects the importance of *Plasmodium falciparum* malaria—the dominant species prevalent in sub-Saharan Africa—as a major etiology of anemia in sub-Saharan Africa. Anemia due to *P. falciparum* infection is likely to be the result of either recurrent infection or the progression of untreated infection to severe malaria. Severe malaria has been shown to present as severe malaria anemia in young children in areas of high malaria incidence[47,48].

The analysis reveals a substantial degree of non-linearity for certain covariates, indicating that the effects on anemia may depend on the levels of these covariates. For example, the slope of the estimated prevalence changes with increasing age, i.e., the effect is non-linear (see the top panel of Fig. 2). Hence, the use of a flexible modeling approach such as structured additive distributional regression allows such associations to be estimated in a data-driven manner, which is critical when analyzing, for example, the complex

interactions between environmental factors, climate change, and anemia prevalence.

By utilizing a distributional regression model to characterize the dispersion of Hb levels, we jointly estimate the prevalence for all forms of anemia using readily available survey data. Creating a unique data set by incorporating remotely sensed information also allows for spatial extrapolation, facilitating the generation of high-resolution estimates for this severe global public health concern. This approach allows our knowledge for the first-ever subnational mapping of any form of childhood anemia in sub-Saharan Africa and South Asia, two regions identified as having persistently high prevalence rates of malnutrition and anemia. The integration of distributional regression and remotely sensed data offers valuable insights into the spatial patterns and severity of anemia in these regions at a high resolution. These insights can help to identify areas where interventions and resources are most urgently needed and can support the improvement of monitoring systems for the SDGs, particularly SDG—*Malnutrition: End all forms of malnutrition*.

Anemia remains a persistent problem within poor households. In addition, targeted policies are necessary to tackle the issue of anemia in impoverished populations effectively to distribute scarce resources effectively. The identification of hotspots emphasizes the geographical

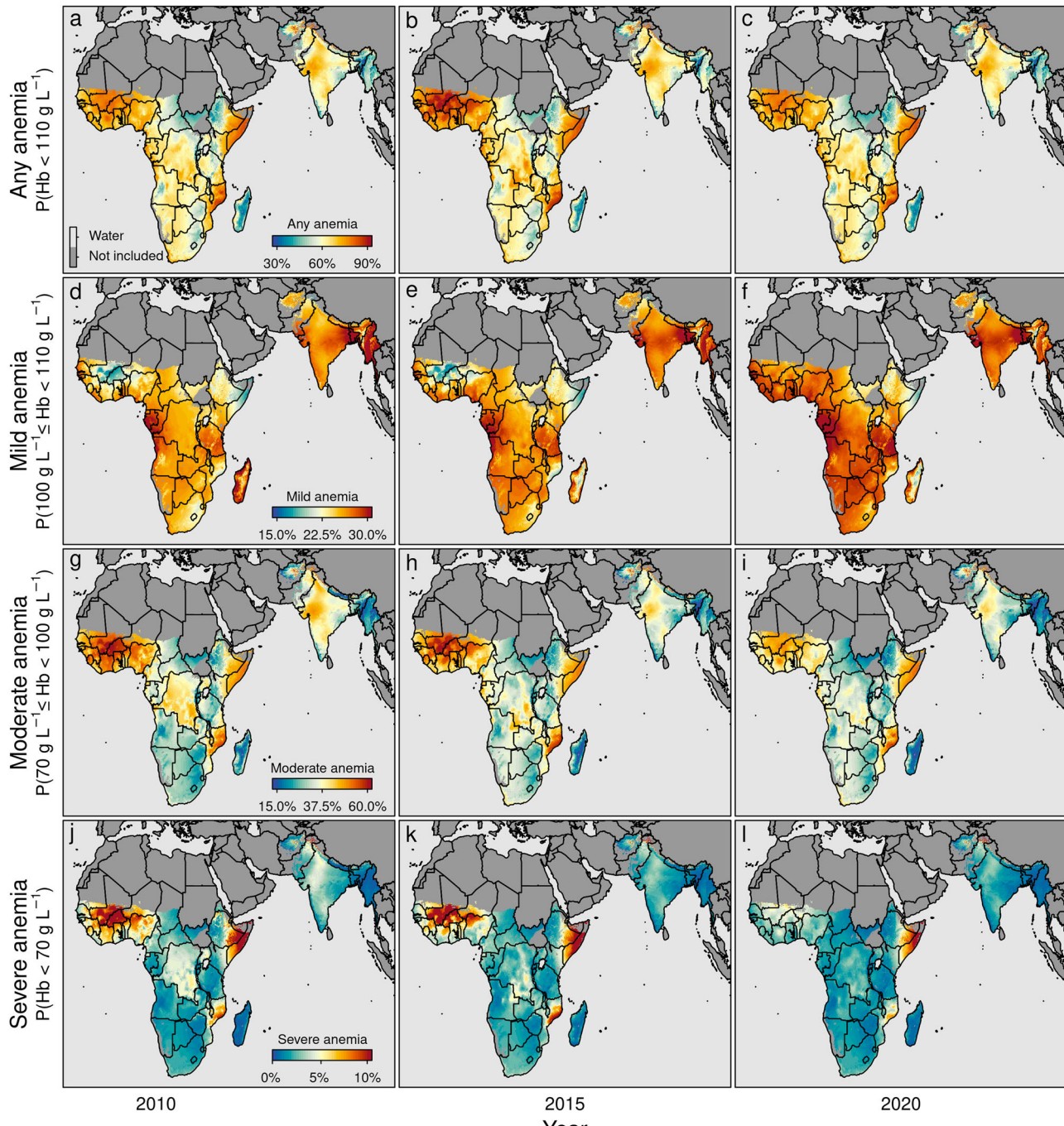

**Fig. 6 | Predicted marginal spatio-temporal prevalence of anemia. a–c** any anemia (i.e., P(Hb < 110 gL$^{-1}$)); **d–f** mild anemia (i.e., P(100 gL$^{-1}$ ≤ Hb < 110 gL$^{-1}$)); **g–i** moderate anemia (i.e., P(70 gL$^{-1}$ ≤ Hb < 100 gL$^{-1}$)); and **j–l** severe anemia (i.e., P(Hb < 70 gL$^{-1}$)). The maps show the estimated anemia prevalence by severity among children aged 6–59 months for the years 2010, 2015, and 2020, respectively. Note that the non-geo-referenced covariates are fixed at the median (metric covariates) or the mode (categorical covariates), respectively. Note that the boundaries reflect administrative boundaries at the country level. Furthermore, note that although the color palettes are identical across rows, the corresponding values differ across rows. In addition, note that pixels categorized as *Barren*, or *Permanent Snow and Ice*, and pixels above 3750 m (1900 m) of altitude in sub-Saharan Africa and South Asia (Madagascar) are flagged as *Not included*.

concentration of anemia prevalence and underscores the importance of targeted interventions and region-specific strategies to address this issue. By focusing efforts on regions with a high burden of anemia, policymakers, and healthcare providers can work toward effective solutions to reduce the prevalence and mitigate the adverse health effects associated with anemia.

While this study sheds light on potential contributing factors to anemia, the observational nature of the analysis precludes drawing causal mechanisms leading to anemia prevalence. Furthermore, this study does not explore specific interventions or policies aimed at addressing the needs of such households. Therefore, further research to identify the affected population and to improve the Hb level is necessary. Moreover, additional research is needed to identify the underlying causes contributing to this discrepancy in anemia prevalence between different regions and populations.

## Data availability

All data used in the analysis of the main text and Supplementary Information are available from the cited sources (see Supplementary Table 1). Supplementary Table 1 also provides URLs and references to the data

sources in the last column of the table. Note that all of the data sources cited in Supplementary Table 1 are freely available from the sources cited, but we do not have permission from ICF International to distribute the DHS data to other researchers. However, DHS survey data are freely available from the cited source upon registration. The underlying data to reproduce Figs. 1–6 of the manuscript is provided in Supplementary Data 1 to Supplementary Data 6. The figures can be plotted using the provided R-script figures. R included in the Supplementary Software.

## Code availability

**Software** The results of this paper have been accomplished using custom software, tailored to be used at the high performance computing (HPC) infrastructure *LEO* of the University of Innsbruck. For that purpose the statistical software R[49] using the following R packages have been used: bamlss[37,50,51], backports[52], broom[53], coda[54], codetools[55], colorspace[56,57], deldir[58], dismo[59], gamlss.dist[60], maps[61], mgcv[62], nlme[63,64], pillar[65], raster[66], rgeos[67], rgdal[68], rnaturalearth[69], rnaturalearthdata[70], rnaturalearthhires[71], scales[72], scoringRules[73], sf[74], smoothr[75], and sp[76,77]. The custom computer code in R[49], which was used to perform the statistical analysis is provided in the Supplementary Software.

**Computational details** To illustrate the approach described in detail in the *Supplementary Methods* and to make our computational results more transparent the R-script example.R illustrates for simulated data the individual steps of the modeling approach. Moreover, all the relevant R-scripts are provided in the Supplementary Software. Please note that the code is custom tailored to the HPC infrastructure *LEO* of the University Innsbruck and maybe adaptions to other systems would be required. In addition, note that we do not have permission from ICF International to pass on the DHS survey data to other researchers.

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

## Acknowledgements
The computational results presented have been achieved (in part) using the HPC infrastructure LEO of the University of Innsbruck. We thank ICF International, Inc. and USAID for conducting the DHS, and providing public access to the data. We are grateful to the three anonymous reviewers for their critical yet constructive feedback that led to substantial improvements in the paper. This research was funded in whole/in part by the Austrian Science Fund (FWF) grant https://doi.org/10.55776/P33941. For open access purposes, the author has applied a CC BY public copyright license to any author-accepted manuscript version arising from this submission.

## Author contributions
Conceptualization and funding acquisition: J.S., K.H., N.U. Methodology and statistical analysis: J.S., N.U., M.W. Visualization: J.S., N.U. Project administration, and supervision: N.U. and J.U. Writing–original draft: all authors.

## Competing interests
The authors declare no competing interests.

## Additional information

**Johannes Seiler** ®[1,2,3] ✉, **Mattias Wetscher** ®[1], **Kenneth Harttgen** ®[4,5], **Jürg Utzinger** ®[6,7] & **Nikolaus Umlauf** ®[1]

[1]Department of Statistics, University of Innsbruck, Innsbruck, Austria. [2]School of Medicine and Health, Technical University of Munich, Munich, Germany. [3]Munich Center of Health Economics and Policy, Munich, Germany. [4]Development Economics Group, ETH Zurich, Zurich, Switzerland. [5]NADEL Center for Development and Cooperation, ETH Zurich, Zurich, Switzerland. [6]Swiss Tropical and Public Health Institute, Allschwil, Switzerland. [7]University of Basel, Basel, Switzerland. ✉e-mail: johannes.seiler@uibk.ac.at

