## [Transparent Peer Review file · Communications Medicine]

High-resolution spatial prediction of anemia risk among children between 6 and 59 months in low- and middle-income

Corresponding Author: Dr Johannes Seiler

Version 0:

Reviewer comments:

Reviewer #1

(Remarks to the Author)
Please see attached file.

[Content of attached file moved across by editor:]

1 Introduction & Summary of the Content

This is a peer-review of “High-resolution spatial prediction of anemia risk among children between 6 and 59 months in low- and middle-income countries” by Seiler, Wetscher, Harttgen, Utzinger, and Umlauf.

2 Summary of the Content

The authors note a need for high resolution (spatial-temporal) estimates of anemia in children to help monitor progress in anemia reduction and target precision interventions. Further, they note that existing approaches of this specific outcome are limited to individual or small selections of countries. There is good reason to want to model larger swathes of countries concurrently: namely this allows the modeler to leverage the entire, larger dataset to model associations between the covariates and anemia that are shared across all countries.

To do this, they employ what they call “full probabilistic Bayesian distributional regression models” to estimate the distribution of hemoglobin (Hb) in children of different ages and sexes across a number of low- and middle-income countries and for every year from 2005 to 2021.

They use flexible distributional models to account for important characteristics (skew) of the Hb distribution that cannot be well-modeled by more often used Normal distributions, and they perform model comparisons to validate the predictive performance of their selected models. They conclude with a brief discussion of some of the results.

3 Overall Feedback

I thank the authors for an interesting which aims to provide needed high-resolution estimates of a high-burden disease. The models that they chose seem well-suited for the task at hand.

That said, there are a number of key points about the modeling and validation process that were opaque to me due to difficult-to-parse sentences pertaining and some lack of clarity in the model and validation description. Further, while the provided validation results look good, I have some questions and concerns about the validation process: namely, if the primary goal is to provide pixel-level estimates, some pixel-level validation is needed.

I believe the paper would benefit greatly from improving the clarity of the model description and the inclusion of additional validation results to help the reader understand and trust the modeling decisions and, ultimately, the estimates.

4 Major Remarks

4.1 Model & Model Validation

1. The authors call the model a “full probabilistic Bayesian distributional regression model,” yet there is no mention of prior distributions in this paper or in [2], which is the reference text for the general approach used in this paper. If this is a Bayesian model, where are the priors? How were they chosen? Does the choice of prior notably impact the final results (see next comment)? Either way, please clarify.

2. If this is a Bayesian model, noting that it contains many covariates and nonlinear effects, how robust are the final outputs to the choice of the priors? If there are priors, please provide explain their appropriateness based on expert input from anemia specialists, or demonstrate that the estimates are robust to the (more arbitrary) choice of prior.

3. In Table S1, some covariates are monthly. Time is not mentioned in the bulleted list on pg36 that provides an overview of the effects (space-time interactions are mentioned, spatial details are mentioned, but nothing extra about time is provided). Then, in Table S3, survey year is described as metric, not continuous. Generally, how is time indexed in the model? Does the model incorporating monthly data, making monthly estimates and then aggregating/averaging to years (monthly covariates suggest this may be possible), does it have an annual discrete time model (survey year as metric suggests this could be), or is it continuous time (thin-plate spline on space-time)?

4. Can the authors clarify their validation procedure?

- An 80-20 test/train split is discussed. Was the training data chosen at random from across the entire dataset? Extra care typically needs to be taken with correlated data because, generally, “non-independence of hold-out data from the training data erroneously makes models appear more reliable than they are, enticing us to have more faith in their predictions than is actually warranted” [1].

- On line 991 it says that the “... model is calibrated on data that have not been used to develop the final full probabilistic distributional model.” Does that mean the testing data was used at an earlier stage of the fitting procedure? (e.g. to select the restricted covariate set included in the final model)? I ask because the authors mention (line 944) that the covariate selection backfitting algorithm is updated using the “out-of-sample log-likelihood of each predictor, η_k .” Which out-of-sample data is being used to evaluate this step? The 20% testing data? Or in this step somehow using the 80% training data less the random subset of the data that is currently being used in the updating step?

- In that vein, please clarify the indices used in equation (3) on pg 39 (is j indexing over possible distributional models?) and the undefined parameter, v .

5. I don't find the validation results shown in Figure 1 to be particularly persuasive (though they do look promising). One of the main purposes of this model is to generate estimates at the pixel-level, which means some validation at the pixel-level should be provided. This can be challenging, but, I think it's necessary to validate the model at the resolution of primary interest. There are any number of ways this could be done. For example, the scatterplots and correlation between the pixel-estimates of Hb and the observed Hb could be shown, or the posterior predictive intervals at each withheld observation location can be checked to see if the observation is inside the interval and these binary outcomes can be averaged over areal units (admin 1, admin 2, etc) if you want to assess the categorical anemia outcomes. Another idea, though not exactly at the pixel level, would be to compare the modeled results aggregated to the admin 2 level against admin 2 level DHS estimates (which I believe should be available or can be calculated).

6. Some of the covariates are (heavily) modeled estimates (e.g. all the Malaria Atlas Project covariates). Is it possible to account for the uncertainty of these covariates in the model? If not, I think it is worth mentioning this as a limitation which will, most likely, lead to an underestimate of the true uncertainty in the outputs.

7. The following are more minor comments, but taken together they make the overall model description difficult to follow, which is a major concern:

- line 846: “Re-estimate sparse model”. This is the first time a sparse model is mentioned in the paper. From looking at the cited modeling references, I believe this means something more like “fit the model with the selected covariates”.

- Line 891: “to allow the ST effects [missing word: to] vary smoothly with the sex of the child...” The meaning of this is unclear to me. Are the outputs somehow smoothing over the binary sex variable? Or is this trying to convey that the ST effects vary smoothly conditional on sex? If so, what about the other categorical covariates?

- line 940: “Non-informative covariates are selected... using the algorithm allows [missing word: one/the model] to select only ... terms ... with the greatest contribution ...” Are the non-informative covariates removed or selected?

- With the described multi-step process, is it possible (or better yet straight-forward) for someone to reproduce these results?

4.2 Anemia

1. Was any attempt made to use the survey DHS provided survey weights? If yes, how? If not, what are the implications of ignoring them?

4.3 Results

1. One of the great advantages of a full probabilistic model is that it quantifies the uncertainty of the generated estimates. I don't believe I saw any results which showed or discussed the uncertainty present in these results. I encourage the authors to provide some visualizations showing (mapping) the uncertainty of the anemia prevalence estimates since the magnitude of this greatly impacts the interpretation of the results. Without understanding the uncertainty of the results, it makes it impossible to assess, for example, if the non-linear impact of temperature on prevalence in India and Madagascar really should be interpreted as a positive association (as described in lines 172-174) or if it is not statistically discernible from a flat line.

2. The results paragraph (lines 212-221) seems quite general and vague. All this wonderful effort has gone into the analysis and a few more specific examples of insights gleaned seems warranted. I believe a more detailed results section would more clearly demonstrate the benefits of this method and what this paper adds to the scientific literature.

3. Somewhat related, the authors allude, multiple times (eg line 58,), to their model's ability to provide "insights into the underlying causes of anemia," but I didn't find much discussion of these, especially given the number of included covariates.

5 Minor Remarks

1. Table S1: under 'Conflict' it states "1 if more than 5 conflict incidents ...", yet in Tables S3 and S4 it states "Indicator whether a conflict...". Is the variable 1 for one or more, or for 5 or more conflicts?

2. Table S4: model term f22 is described as spatio-temporal but it has no time index. Perhaps you meant spatial-age effect?

3. Lines 131-136: there are some grammatical errors which make this sentence hard to parse. In addition, this is the first time the "naive models" are mentioned, making it difficult/impossible for the reader to correctly interpret these validation metrics.

4. I've already pointed out a few minor corrections in the above quotes. Here's a few others I noticed:

- Line 136: Similar → Similarly.
- Line 900: missing section number. (I think I saw another missing section number somewhere in the appendix, but I didn't note where.)
- The title to Table S3 is hard to parse with multiple "included"s and "model"s.

3

References

- [1] David R. Roberts et al. "Cross-validation strategies for data with temporal, spatial, hierarchical, or phylogenetic structure." In: *Ecography* 40.8 (2017), pp. 913–929. doi: 10.1111/ecog.02881.
- [2] Nikolaus Umlauf and Thomas Kneib. "A primer on Bayesian distributional regression." In: *Statistical Modelling* 18.3-4 (2018), pp. 219–247.

Reviewer #2

(Remarks to the Author)

General comments

Congratulations to the team, a very well built and written paper. It provides interesting insights on anaemia. The steps followed to understand the data, distributions, selecting covariates and model building to validation is commendable. The analytical methods are well elaborated.

Minor comments that the authors need to consider to make some sections clearer, revisit the texts for a more focused narratives and correct few mistakes.

Specific comments

- Abstract: I suggest to include some of the main insights from the work in the abstract if possible – at the moment it weights too much on data and methods.
- In all these sections - distribution of Hb, Probabilistic distributional regression, Variable selection, Calibration and convergence checks, etc - some details could be omitted to make the narratives precise and succinct. Lengthy descriptions make the paper too long unnecessarily. E.g., stating training and test data should be simplified as that doesn't need too much elaboration or could just add a reference for interested readers.
- Understanding the regional disconnectedness, and the approach used by the author to model the regions separately, however, that may bring a query on input data – to clear this, I would suggest including in the Supp materials maps indicating locations of the observed PSU for SSA, Madagascar and South Asia.
- Lines 172-176: The statement "This finding is particularly concerning, as rising temperatures due to climate change could potentially exacerbate the prevalence of anemia in sub-Saharan Africa" seems like an over emphasis on the effects of climate on Anemia in SSA.
- What was the basis and criteria for presenting results of selected age categories Eg in Figure 4? Why not blocks of age say 6– 18m, 18-30m, etc? It is a bit hard to use a-month-specific results for practical recommendations, in my opinion.
- Lines 200 onwards: Fair to weigh the descriptions a lot on the problematic areas, increasing trends observed, etc, but would be also great to speak about the reduction observed in some countries both within and outside SSA. This is done but I think on a very minimal level.
- I would present results of both boys and girls as main figures in the manuscript or as Supp material. Separating them

creates ambiguities to comprehend what is going on. If the figures are too many you could opt to only present specific categories.

- The subsection on “Parasites”: The section described important variables for existence of Hookworm larvae, but not clearly stating what variables were used as a proxy for it in the analysis. Kindly check and ensure clarity.
- One of the common analytical approaches to model anemia data by these classes is a multinomial (logistic) model. Authors described (pros/cons) various approaches but not this one – I am curious to know why?

Editorials

Figure 2:

- As this is among your main Figures for the manuscript, the caption should include explanation of the abbreviations/ISO3 used – this may not be trivial to all readers.
- Line 161 – with the observed trends I wouldn't say this is highly negatively correlated, could the authors elaborate more here? What measure has been used to come with that quantifiable conclusion? The trends observed by age are steeper than the ones for HHD wealth – surprisingly this was not mentioned.

Footnote 5:

- Not sure I understood this part - as at birth and directly after birth the Hb levels are usually higher and distort possibly the estimates. Could the authors elaborate this further/ make it clear.

Fig. S1

- The red/green dots are not really visible. Check for alternative colours

From line 177 - Spatio-temporal dynamics

- Authors should revisit this section and check that the narratives are correct, referencing of Figures is not confusing, and all the captions are correct – for example Fig 4 is referred to as for girls in line 183 and as for young children in line 197; the caption for Fig 4 mentioned the results are for 12 months but the figure shows for 6months, etc

Reviewer #3

(Remarks to the Author)

Summary:

The manuscript “High-resolution spatial prediction of anemia risk among children between 6 and 59 months in low and middle-income countries” uses a fully Bayesian probabilistic distributional regression modelling framework to estimate the prevalence of mild, moderate and severe anaemia in children under five years of age in sub-Saharan Africa and South Asia. The model used anaemia measurements from DHS surveys and a selection of spatial-temporal covariates such as temperature, land coverage and malaria prevalence to predict the prevalence of anaemia at a 20x20km spatial resolution.

Estimates of the prevalence of mild, moderate, severe and any anaemia are presented for children at different ages for 2010, 2015 and 2020. They highlight anaemia being highest in young children (6 months) and show little temporal changes in burden. The highest levels of anaemia were identified in western sub-Saharan Africa and parts of east Africa. This presents the first subnational estimates of childhood anaemia in these regions and may prove a useful tool for benchmarking and decision making to tackle anaemia and food insecurities (SDG2) in children.

General comments to author.

Overall, I found this an interesting, well conducted study. The modelling strategy appears well reasoned and appropriate to the data and the figures presenting the results are clear and informative. However, I think that it could benefit from a more structured methods section and a more quantitative assessment of the results. Specifically:

- 1) I found myself having to go to the supplementary materials to follow the methods. I think that it would be beneficial to have a brief overview of the input data and the modelling framework at the beginning of the methods section and then go into brief details of the probabilistic regression model, distribution choice, variable selection, model calibration/convergence. Whilst I assumed the DHS anaemia measurements were used as input data I had to go to the supplement to confirm this. More balance between the main text and supplementary methods is required.
- 2) I think that the results could be strengthened and made more impactful by including the a summary of the estimates of the prevalence of anaemia from your model. For example, I would like to see the overall prevalence and total number of children estimated to have mild/moderate/severe anaemia in 2010 and 2020 in your study area (mean and uncertainty intervals) as well as the values for countries or locations you identify as hot spots. This would help to put the results into context and enable comparisons to previous estimates.
- 3) One of the strengths of this study is it being a subnational analysis. It would be interesting to assess the in-country variation in anaemia prevalence to highlight inequalities in determinants of anaemia such as food security and infectious disease prevalence. Using metrics such as the Gini coefficient or calculating the mean relative deviation are potential methods which I would like to see explored. Additionally, it would be interesting to analyse the data by urban/rural classification to look at the disparities there.
- 4) Did you find data sparsity and issue as you are estimating for many countries with no data. Did you test the model holding out whole countries worth of data to assess the model performance here?
- 5) In the discussion I think that it is important to compare you're results to previous estimates of anaemia burden. How so these compare to the likes Stevens et al (doi: 10.1016/S2214-109X(22)00084-5. Ref 34 in this manuscript).

Specific Comments to author:

Introduction

Lines 37-40: You state that studies on the determinants of anaemia and therefore knowledge of the trends and burden across LMICs is limited. I think here it is worth noting the estimates from Stevens et al (doi: 10.1016/S2214-109X(22)00084-

5. Ref 34 in this manuscript) as they provide national estimates of anaemia burden in children u5 for 2000-2019, giving you another estimate of anaemia across LMICs.

Line 57-60: you list variables your study will provide insights into for their contribution to the causes of anaemia, however, these aren't included as covariates in the model so this is an inaccurate statement.

Methods

Measurement of anemia

Line 68-69: To be pedantic, the definition of anaemia is $<110\text{g/L}$ therefore non anaemic would be greater than or equal to 110g/L (could reword to make this consistent with other locations in your manuscript).

Need to link this section to the DHS surveys and clarify this is where you are getting your measurements of anaemia from. Are you using the adjusted or unadjusted Hg levels from DHS?

Distribution of Hb levels

Lines 73-93: An overview of the modelling strategy before going into the details of the data distribution would be beneficial and put this in context. I think that only a brief description of the Hb level distribution and the importance of selecting an appropriate one (a couple of sentences) is required here and most of this section would be better suited to the supplement.

Probabilistic distributional regression

Line 142-143: the correlation between input data and prediction at the country level is very good. But as this is a subnational model, how do the subnational comparisons look (e.g. at administrative division 1)?

I would like to see a bit more of a description of the model here, including what package and software were used to fit it.

Please include a location where the code used to fit the model is shared.

Results

Effect of covariates (marginal probabilities)

Line 173: Clarify in the text here that the "positive association" you are referring to is with temperature. I think discussion of the strength of this association is also important as it seems quite a weak association in India.

Line 174-6: The conclusion that climate change is going to potentially exacerbate anaemia in sSA quite a leap from these results, when the issue is far more complex with temperature/rain fall/crop yields impacting food security and other confounding factors. I think this is something more suited to being expanded on in the discussion, with the inclusion of other referenced evidence.

Spatio-temporal dynamics

Line 185-8: This is a very interesting result. Why may this be the case? Are other determinants of anaemia increasing if malnutrition is decreasing (malaria burden also decreases in this time frame but what about schistosomiasis etc)? I think this warrants further attention in the discussion section.

Line 190: "Many countries showed a(n) increase in the prevalence level of anaemia" – I would like to see some detail on this, how many countries saw an increase, which countries increased the most, what was the level of increase.

Corresponding numbers from the modelled estimates to back up this assertion would strengthen the results.

Line 191-193: I don't think this statement of a global comparison is required here, I would prefer more information on your results here and to compare this to other estimates in the discussion.

Line 194: 'Substantial burden' – what was the burden? How many cases of anaemia in 2020 does your model estimate?

Line 196: states that you have identified hot spots. You repeat this assertion with a bit more info in lines 202-206 – not needed in both places. I would like to see some numbers to accompany your observation of 'hotspots'. How high does the prevalence of anaemia reach in these locations? It would also be interesting to note and compare this to cold spots.

Line 197-198: Again, having some numbers from your model would be good to quantify this statement.

Line 214-216: Again, some numbers would be good here.

Fig 3: Could the same colour ramp be used as in figs 4 & 5?

Fig 4 & 5: Visually there appears to be little difference between the maps for girls and boys. This is the first mention of the analysis being stratified by sex and it isn't discussed further. If there are no noteworthy differences between the boys and girls to be discussed could these figures be shown for the sexes combined (and then specify differences in the supplement)?

Fig 4: It looks like there is a decrease in anaemia in western sSA for the 59 months map in 2020. This is contrary to what you have said in the text, is this an important result?

Discussion

Line 230-233: Yes, the predictive performance of the model at country level is impressive but as what is novel about this study is the subnational estimates it would be interesting to assess the performance sub-nationally too. How did you select your held out data, was this random or stratified over time and location? Unless whole countries or surveys have been held out I would expect the country level correlation to be high as there is a wealth of other point level data from that country-year included in the model. Can you describe this model validation a bit further (not required in main text but in the supplement this could provide strength to your assertion that your model has excellent predictive performance)

Line 240-242: "This choice reflects the fact that most fatal cases and infections are caused by the dominant species *P. falciparum*, which is prevalent in sub-Saharan Africa." I don't think this is an accurate assessment of what the inclusion of malaria incidence in the model reflects, especially as you are not looking at fatalities here. Rather it highlights the importance of Pf malaria as an aetiology of anaemia in these regions. Anaemia due to Pf will likely be due to either repeat infections or untreated infections progressing to severe malaria. Evidence suggests that in high Pf locations severe malaria presents in children at a young age as severe malarial anaemia, which may be reflected in your results. I think this is an important point that warrants discussion.

Line 246-248: I do not follow how non-linearity with covariates indicates that the effect on anaemia is dependant on those covariates, surely dependence is not subject to the shape of the association and the is the comparison of the model performance with and without the covariates not a better indicator of this? You say non-linear associations improved the model, did you fit it enforcing linear relationships or is this an assumption?

Line 258-260: how do your results compare to Stevens et al (doi: 10.1016/S2214-109X(22)00084-5. Ref 34 in this manuscript) who produce national estimates of childhood anaemia. Please clarify that these are the first subnational estimates of anaemia for these regions, not the first.

Line 264-5: How would these estimates contribute to an early warning system of food insecurity? Whilst I agree this is a good tool for identifying areas requiring support, it is more a long term tool and useful for predictions in areas where measurements have not been done/are not possible. I don't think it works as an early warning system as (a) the food insecurity will be coming before the anaemia, and (b) the input data of anaemia measurement are time consuming and expensive to undertake therefore not regularly updates, and (c) for an early warning system you would be looking for sudden changes in your estimates, if you don't have the input data regularly updated than this would be reliant on your covariates which would need to accurately reflect shocks; the included covariates appear annual at best.

Version 1:

Reviewer comments:

Reviewer #1

(Remarks to the Author)

1 Introduction & Summary of the Content

This is a peer-review of the revised "High-resolution spatial prediction of anemia risk among children between 6 and 59 months in low- and middle-income countries" by Seiler, Wetscher, Harttgen, Utzinger, and Umlauf, submitted Aug 2024.

2 Summary of the Content

The authors note a need for high resolution (spatial-temporal) estimates of anemia in children to help monitor progress in anemia reduction and target precision interventions. Further, they note that existing approaches of this specific outcome are limited to individual or small selections of countries. There is good reason to want to model larger swathes of countries concurrently: namely this allows the modeler to leverage the entire, larger dataset to model associations between the covariates and anemia that are shared across all countries.

To do this, they employ "full probabilistic Bayesian distributional regression models" to estimate the distribution of hemoglobin (Hb) in children of different ages and sexes across a number of low and middle-income countries and for every year from 2005 to 2021. They use flexible distributional models to account for important characteristics (skew) of the Hb distribution that cannot be well-modeled by more often used Normal distributions, and they perform model comparisons to validate the predictive performance of their selected models. They conclude with a brief discussion of some of the results.

3 Overall Revision Feedback

I thank the authors for working so diligently to address my comments and those from the other reviewers. It looks as if it took substantial effort, but I agree with their assessment that the paper is now much improved. Congratulations. I would like to reiterate two small (but I believe important) requests from my initial set of feedback which were not completely addressed in the revision.

4 Major Remarks

My line numbers reference the combined marked up pdf document.

1. I appreciate the extra modeling clarity that the authors have included in the revision. I still have some questions about the priors used in the modeling framework and I think it is important that the authors completely specify their model choice, including specifying the details of their priors.

a) Priors are now mentioned in the supplement and their first mention appears at line 1065 under the subsection "Re-estimate the sparse model." Is a full Bayesian model not used in the initial covariate selection model fits? If not, how do those models differ from the final model and what are the implications of using different models during variable selection vs the final model?

b) Line 1065 says "a common choice of priors for linear effects are uninformative normal priors for γ , ie $p_{jk}(\gamma|k|y) \propto \text{const.}$ " I have a few concerns with this.

i. First, I would not call a flat/constant prior normal.

ii. Second, "flat" priors are not synonymous with uninformative priors (nor are they invariant to parameter transformations). The common example is that a flat prior on a standard deviation from a Gaussian distribution, σ , puts a lot of prior weight of σ being far from zero, ie it is actually a strong prior on having a large value of σ .

iii. Third, constant priors are not proper distributions and there is no guarantee that the posteriors will be proper. This seems at odds with the authors' claims (in both the rebuttal letter and on line 1069) that "the primary reason we use a Bayesian model is not to incorporate prior knowledge to improve the model but to achieve valid inference." I understand the sentiment of their comment (in fact, I often use Bayesian models for the same reason), but if valid inference is the primary reason for employing Bayesian methods, then why not use proper Bayesian priors?

There are some real reasons to be hesitant about using improper priors (eg see Marginalization Paradoxes in Bayesian and

Structural Inference by Dawid, Stone, and Zidek.)

c) Line 1067 says that, "For non-linear effects, which are typically based on a basis function approach, a multivariate normal prior for β_{lk} , with a model term-specific precision matrix is used." The paragraph concludes with "generic multivariate normal priors for smooth functions have demonstrated 1072 high efficiency and robustness in numerous applications". Do the priors used on the non-linear effects in this paper have "model term-specific precision matrices" or are "generic multivariate normal priors" used? Either way, please write down the specific choice of priors used. As my previous comments about the flat prior attempt to convey, the details of the priors

used do matter. This is even more important since the complete code for the full model (understandably) is not available to reproduce the results and we have no other way of knowing how the model was completely specified.

2. My other comment still pertains to validation and model presentation. The new clarifications around the validation method and the admin2 level validation results go quite a long way towards assuaging my concerns. Still, I feel there is mismatch between the model resolution and presentation (continuous/pixel) and the validation (discrete admin units). You have a number of good options to resolve this completely, and all pertain to aligning these two concepts or taking a few sentences to explain why they are different.

- You could add validation results to the supplement which show how well the model is performing at the continuous/data observation level.

- If the primary target of inference is admin2 units, then that should be clarified and in this case I think it is also worth discussing why you are using continuous spatial models for discrete spatial outcomes.

- You could show your main continental results (Figure 4) at the admin 2 level and not at the pixel level.

- etc

Reviewer #2

(Remarks to the Author)

I went through the Authors' responses on the comments I provided and those raised by other reviewers, reviewed the revised manuscript and all the additions done in the main document and the Supp File. Extensive revision and improvement have been done into the paper.

Specifically:

- The additions in the Abstract, Methods [modelling and calibration] and Results (both in the main manuscript and Supp file) have created clarity in the areas that were ambiguous or hard to follow. I appreciate additions of Country-specific and subnational results in the paper.

- Revisions done to the Figures and Plots is appreciated

- Revisions in the interpretation of results is sufficiently done

- If Authors wish - In the Discussion - There was no need to start by mentioning the Limitations of this study, that is a great addition but still can go at the end of the section- before Conclusion.

No further comments for this work.

Reviewer #3

(Remarks to the Author)

1) Overview

I thank the authors for the revisions to the manuscript "High-resolution spatial prediction of anaemia risk among children between 6 and 59 months in low-and middle-income countries". The revision have addressed my initial comments well and greatly improved the manuscript, both in terms of readability, strength of the results and utility/interpretation of the results. The addition of Figure 3, extension of the discussion of the results and model validation are particularly beneficial.

2) Specific comments:

I just have a couple of remaining points/queries that I think would be beneficial to address.

The modelled estimates in some countries appear to have changed quite a lot from the previous submission. The countries I have concerns about have very different estimates than in WHO and appear are a bit inconsistent with my preconceptions of general health levels in these countries. These countries have no data from DHS so I would like to understand what is driving these estimates:

a. Somalia is estimated to have the highest prevalence of anaemia of all countries.

b. South Sudan is estimated to have the lowest prevalence of any anaemia. This especially shocked my as you have discussed the importance of household wealth as a covariate in the model

c. Finally CAR is also a considerably lower anaemia prevalence than I would expect (again, based on HH wealth being an important factor)

Minor comments:

Line 174: Tables S4-7 are the nationally aggregated estimated of anaemia, not the pixel level estimates as stated.

Line 203: States tables s4-7 are for years 2020, 2019 and 2020 (typo for first 2020 = 2010).

Line 217: "194million children" – great to have this estimate added, can you add the uncertainty?

Line 295-299: A good start in talking about the differences but as I mentioned, Somalia has no surveys and is coming up with very high levels of anaemia. I am not disputing that this could be true, and I am actually more concerned about the low prevalence of anaemia your model estimates in South Sudan and CAR. Are there any sources which can confirm these patterns? I think this manuscript could benefit from a bit more of a discussion about the disparities between yours and WHO estimates and what is driving them.

Version 2:

Reviewer comments:

Reviewer #1

(Remarks to the Author)

1 Introduction & Summary of the Content

This is a peer-review of the second revision of "High-resolution spatial prediction of anemia risk among children between 6 and 59 months in low- and middle-income countries" by Seiler, Wetscher, Harttgen, Utzinger, and Umlauf, submitted Nov 2024.

2 Summary of the Content

The authors note a need for high resolution (spatial-temporal) estimates of anemia in children to help monitor progress in anemia reduction and target precision interventions. Further, they note that existing approaches of this specific outcome are limited to individual or small selections of countries. There is good reason to want to model larger swathes of countries concurrently: namely this allows the modeler to leverage the entire, larger dataset to model associations between the covariates and anemia that are shared across all countries.

To do this, they employ Bayesian distributional regression models to estimate the distribution of hemoglobin (Hb) in children of different ages and sexes across a number of low- and middle-income countries and for every year from 2005 to 2021. They use a batchwise backfitting algorithm to select covariates, use flexible distributional models to account for important characteristics (skew) of the Hb distribution that cannot be well-modeled by more often used Normal distributions, and they perform model comparisons to validate the predictive performance of their selected models.

They conclude with a brief discussion of some of the results.

3 Overall Revision Feedback

I again thank the authors for working to address my comments and those from the other reviewers. All my main concerns have been addressed and I would recommend that the paper be accepted.

4 Minor Remarks

I agree that your pixel-level validation, both in- and out-of-sample, suggest a good model fit for your data and suggest that it is capable of reliable predictions in similar out-of-sample settings. I only have one question for you: do you think it makes sense to show the credible intervals for the pixel estimates (as you've done) or to include the posterior predictive credible intervals? I can see arguments either way. I think it's clear from the pixel-level validation plot that the 95% of the pixel-level estimates do not contain the DHS reported data observations – but that's to be expected. On the other hand, approximately 95% of the out-of-sample 95% posterior predictive credible intervals should contain the reported data observations. The post. pred. credible intervals will necessarily be wider than the credible intervals for the pixel estimates, and it may make the plots harder to interpret. I would personally be curious to know what percent of your post. pred. credible intervals cover the data, and it may be a useful validation metric for you to check and include, but I leave that to your discretion.

I've enjoyed reading your paper. Thank you for this contribution to precision global health.

Sincerely,
Aaron Osgood-Zimmerman
Assistant Professor of Statistics
Department of Mathematics
Bucknell University

Reviewer #3

(Remarks to the Author)

I would like to thank the authors for their thorough response to my previous comments and their revisions to the manuscript. This has now addressed all of my concerns and I believe has strengthened the manuscript substantially. I think

this is an informative and interesting piece of work that adds value to the field. I have no further comments requiring attention.

Rebuttal Letter to *Comments Raised by the Reviewers* for the Revised Article
*High-resolution spatial prediction of anemia risk among children between 6 and
59 months in low-and middle-income countries (COMMSMED-24-0035-T)*

submitted to

Communications Medicine

August 5, 2024

Comments Reviewer # 1

1 Introduction & Summary of the Content

This is a peer review of “High-resolution spatial prediction of anemia risk among children between 6 and 59 months in low- and middle-income countries” by Seiler, Wetscher, Harttgen, Utzinger, and Umlauf.

2 Summary of the Content

The authors note a need for high-resolution (spatial-temporal) estimates of anemia in children to help monitor progress in anemia reduction and target precision interventions. Further, they note that existing approaches of this specific outcome are limited to individual or small selections of countries. There is good reason to want to model larger swathes of countries concurrently: namely this allows the modeler to leverage the entire, larger dataset to model associations between the covariates and anemia that are shared across all countries.

To do this, they employ what they call “full probabilistic Bayesian distributional regression models” to estimate the distribution of hemoglobin (Hb) in children of different ages and sexes across a number of low- and middle-income countries and for every year from 2005 to 2021.

They use flexible distributional models to account for important characteristics (skew) of the Hb distribution that cannot be well-modeled by more often used Normal distributions, and they perform model comparisons to validate the predictive performance of their selected models. They conclude with a brief discussion of some of the results.

3 Overall Feedback

I thank the authors for an interesting which aims to provide needed high-resolution estimates of a high-burden disease. The models that they chose seem well-suited for the task at hand.

That said, there are a number of key points about the modeling and validation process that were opaque to me due to difficult-to-parse sentences pertaining and some lack of clarity in the model and validation description. Further, while the provided validation results look good, I have some questions and concerns about the validation process: namely, if the primary goal is to provide pixel-level estimates, some pixel-level validation is needed.

I believe the paper would benefit greatly from improving the clarity of the model description and the inclusion of additional validation results to help the reader understand and trust the modeling decisions and, ultimately, the estimates.

Response We thank Reviewer # 1 very much indeed for the critical but positive overall assessment of our research and the correct summary of our manuscript entitled *High-resolution spatial prediction of anemia risk among children between 6 and 59 months in low-and middle-income countries (COMMSMED-24-0035-T)*. Please find below our point-by-point response to the aspects and questions raised by you and two other anonymous reviewers.

4 Major Remarks

4.1 Model & Model Validation

1. The authors call the model a “full probabilistic Bayesian distributional regression model,” yet there is no mention of prior distributions in this paper or in [1], which is the reference text for the general approach used in this paper. If this is a Bayesian model, where are the priors? How were they chosen? Does the choice of prior notably impact the final results (see next comment)? Either way, please clarify.

Response Thank you for pointing this out. The primary reason why we use a Bayesian model is not to incorporate prior knowledge to improve the model but to achieve valid inference, which is difficult and sometimes even impossible to obtain for distributional regression models using the frequentist approach. In this context, a common choice is to use generic multivariate normal priors, which have demonstrated high efficiency and robustness in numerous applications (see, e.g., [1–5]). We now added additional information in the supplementary information on this. The primary reference is [6]; we have updated this accordingly.

2. If this is a Bayesian model, noting that it contains many covariates and nonlinear effects, how robust are the final outputs to the choice of the priors? If there are priors, please provide explain their appropriateness based on expert input from anemia specialists, or demonstrate that the estimates are robust to the (more arbitrary) choice of prior.

Response As mentioned in the response of the previous comment, we use a Bayesian model to obtain valid inferences for distributional regression models. In this context, priors are typically uninformative and based on multivariate normal kernels. We hope that the additional information and further references provided in the supplementary materials clarify this approach.

3. In Table S1, some covariates are monthly. Time is not mentioned in the bulleted list on pg36 that provides an overview of the effects (space-time interactions are mentioned, spatial details are mentioned, but nothing extra about time is provided). Then, in Table S3, survey year is described as metric, not continuous. Generally, how is time indexed in the model? Does the model incorporating monthly data, making monthly estimates and then aggregating/averaging to years (monthly covariates suggest this may be possible), does it have an annual discrete time model (survey year as metric suggests this could be), or is it continuous time (thin-plate spline on space-time)?

Response We apologize for this inconsistency in the presentation of the preprocessing steps of the data set. We have now added a detailed flowchart (Fig. S 1 in the Supplementary Materials) that summarizes how the final data set was assembled and how we accounted for different periodicities and spatial resolutions. In addition, we now include Equation (3) in the Supplementary Materials, which is the full predictor used in the selection step to identify informative covariates.

In the model, the survey year is included in terms f_{18} , f_{24} , and f_{25} which build the complex space-time interaction based on a tensor product interaction using thin plate regression splines as the marginal basis. Furthermore, term f_{25} allows the space-time interaction to vary smoothly with the sex of the child.

4. Can the authors clarify their validation procedure?

Response We are happy to provide more details on the validation procedure in the Supplementary Materials of the manuscript. Briefly, the validation procedure can be summarized as follows: First, the input data are divided into training and test data sets, taking into account the geographic dependency structure. Second, we perform a model estimation routine (as outlined in Section *S 3 Methodology* of the Supplementary Materials) to obtain the final models that omit uninformative covariates. Third, we estimate the final models. Fourth, model evaluation, calibration checks, model diagnostics, and cross-validation procedures are done as described in Section *S 3 Methodology* of the Supplementary Materials. In addition, please consider our response to your more detailed questions below.

- An 80-20 test/train split is discussed. Was the training data chosen at random from across the entire dataset? Extra care typically needs to be taken with correlated data because, generally, “non-independence of hold-out data from the training data erroneously makes models appear more reliable than they are, enticing us to have more faith in their predictions than is actually warranted” [7].

Response Thank you for raising this clarification issue. We are aware that random sampling can be problematic in this case. As now shown in the flowchart included in the Supplementary Materials as *Figure S 1*, we accounted for this by following the procedure described by Gething and colleagues [8], which is based on Delaunay triangulation, where we used $\sqrt{\text{area}_i}$ as sampling probability. To explore the consequences of neglecting this important aspect, we also ran the models with the training and validation data split randomly. Between the two methods, the results are quite similar – due to the large number of observations. In addition, to validate this approach, we validated our results using cross-validation, where we ran the final model by country, and for each run, the observations of the training data pertaining to country_{*i*} were held out. Then, the test data – which include observations from the omitted country – were used to assess the effect of omitting country_{*i*} on the estimates. See Section *S 3 Methodology* of the Supplementary Materials for details.

- On line 991 it says that the “... model is calibrated on data that have not been used to develop the final full probabilistic distributional model.” Does that mean the testing data was used at an earlier stage of the fitting procedure? (e.g., to select the restricted covariate set included in the final model)? I ask because the authors mention (line 944) that the covariate selection backfitting algorithm is updated using the “out-of-sample log-likelihood of each predictor, η_k .” Which out-of-sample data is being used to evaluate this step? The 20% testing data? Or in this step somehow using the 80% training data less the random subset of the data that is currently being used in the updating step?

Response Thank you for spotting this poor choice of words on our side. The test data were used only to validate the estimates of our models and were not used in any other step of the estimation process. Please see the flowchart shown in *Figure S 1* that schematically illustrates all the preprocessing steps we carried out. We corrected the first one as follows:

... model is calibrated on data that have not been used in both steps (variable selection and refitting) of the development of the final full probabilistic distributional model ...

The second sentence was unfortunately also misleading: “...out-of-sample log-likelihood of each predictor, η_k .” We reworded the corresponding paragraph and added more details:

In contrast to the classical backfitting algorithm, in this novel variant [9], instead of using in each iteration the complete training data set, the evaluation of the model terms in each updating step is based on a random subset of the data, which is different in each step. Accordingly, the training data set is partitioned into $T = 500$ randomly chosen batches $\mathbf{b}_1, \dots, \mathbf{b}_T$, where samples may overlap (i.e., sampling with replacement), and in each step $t = 1, \dots, T$ the backfitting algorithm computes potential updates on the batch $\mathbf{i} = \mathbf{b}_t$ and selects the best update corresponding to a single model term according to the best log-likelihood improvement on the next batch $\hat{\mathbf{i}} = \mathbf{b}_{t+1}$ (if $t = T$, then the next batch is the first). Using two batches per iteration gives additional stability because the second batch $\hat{\mathbf{i}}$ emulates out-of-sample data. ...

The out-of-sample log-likelihood was actually the log-likelihood improvement on the next batch. Using two batches for the updating, one for the computation of potential updates and one for the selection, helps with improving the stability of the selection algorithm. Please note that there was no test data used in the selection or refitting of the algorithm.

- In that vein, please clarify the indices used in equation (3) on pg 39 (is j indexing over possible distributional models?) and the undefined parameter, ν .

Response Thank you for spotting this. We added better descriptions and more information on the variable selection step and the corresponding Equation (3) which is now Equation (4). You can find in the revised version of the paper that $\nu = 0.1$ is a step length parameter for the update and j refers to the selected effect.

5. I don't find the validation results shown in Figure 1 to be particularly persuasive (though they do look promising). One of the main purposes of this model is to generate estimates at the pixel level, which means some validation at the pixel level should be provided. This can be challenging, but, I think it's necessary to validate the model at the resolution of primary interest. There are any number of ways this could be done. For example, the scatterplots and correlation between the pixel estimates of Hb and the observed Hb could be shown, or the posterior predictive intervals at each withheld observation location can be checked to see if the observation is inside the interval and these binary outcomes can be averaged over areal units (admin 1, admin 2, etc) if you want to assess the categorical anemia outcomes. Another idea, though not exactly at the pixel level, would be to compare the modeled results aggregated to the admin 2 level against admin 2 level DHS estimates (which I believe should be available or can be calculated).

Response Thank you for pointing out the lack of clarity in our validation strategy. This was also pointed out by the other reviewers. To clarify this, we now compare our estimates to the observed descriptive estimates obtained from the DHS data at the admin-2 level. Overall, even at this disaggregated level, we observe a high degree of overlap between our estimates and the descriptive estimates calculated from the DHS data. See also Figure 1 and Figure 2, which plot the predicted prevalence of all forms (i.e., any, mild, moderate, and severe) of anemia against the reported prevalence, and the predicted hemoglobin levels against the observed hemoglobin levels. Overall, the correlation is very high, highlighting the validity of our estimates. Our model performs worst for mild anemia. However, this is not, surprising given the narrow cut-offs (i.e., $P(100 \text{ g L}^{-1} \leq \text{Hb} < 110 \text{ g L}^{-1})$) used to define mild anemia. Note also that since we also predict for countries with no data, and to address potential sparseness issues, we use a cross-validation procedure, see paragraph *Robustness check for potential data sparseness issues – leave one administrative region out cross-validation* of Section S 3 in the Supplementary Materials for details, that further strengthen the validity of our estimates.

6. Some of the covariates are (heavily) modeled estimates (e.g., all the Malaria Atlas Project covariates). Is it possible to account for the uncertainty of these covariates in the model? If not, I think it is worth mentioning this as a limitation which will, most likely, lead to an underestimate of the true uncertainty in the outputs.

Response Reviewer # 1 is correct with this observation. Unfortunately to our knowledge, measurement error correction for a generalized additive model for location, scale, and shape (GAMLSS) is not available thus accounting for the uncertainty introduced by these (heavily) modeled estimates is not feasible. The first version of our manuscript did not discuss any possible and hypothetical limitations. Therefore, in the revised version of the manuscript, we included a separate paragraph in the discussion section where we mention and discuss the limitations of our study. This paragraph reads as follows:

Before discussing our main findings and placing them in a broader context, we also want to address and mention some limitations of our study. First, the measured Hb levels may have been measured or reported erroneously. This may be due to malfunction, or misuse of measurement equipment or

Fig. 1: Scatter plot of the survey based admin-2 level prevalence of any anemia ($P(\text{Hb} < 110 \text{ g L}^{-1})$), mild anemia ($P(100 \text{ g L}^{-1} \leq \text{Hb} < 110 \text{ g L}^{-1})$), moderate anemia ($P(70 \text{ g L}^{-1} \leq \text{Hb} < 100 \text{ g L}^{-1})$), and severe anemia ($P(\text{Hb} < 70 \text{ g L}^{-1})$) reported by DHS and the model based estimates. In addition, the correlation coefficient is reported.

Fig. 2: Scatter plot of the survey-based Hb concentrations reported by DHS and the model-based estimates aggregated to admin-2 regions within countries. In addition, the panel includes 95% credible intervals for each location and the correlation coefficient for each geographic region.

transmission errors. Even though the DHS are known for their high standards of data quality, this point cannot be ruled out. Second, hypothetically, an individual eligible for Hb measurement who normally resides above 1,000 m above sea level was below 1,000 m above sea level in the weeks before the measurement (or vice versa), which could potentially distort the measured Hb level. Third, the uncertainty (see e.g., *Figures 2* and *4*) is particularly high in areas where data are scarce, in areas that are not densely populated (e.g., desert-like areas in South Africa and Namibia), or where the sample size is small (e.g., Madagascar); this usually also underscores the point that newer and updated survey statistics in these regions are urgently needed to closely monitor the progress towards the SDGs in these areas. Fourth, although we have taken great care in the pre-processing steps to merge the data sources, some of the input data are pre-existing (highly modeled) estimates (e.g., malaria prevalence), which are themselves subject to inaccuracies. To our knowledge, there are yet no methods to account for measurement error in GAMLSS models, so the estimated uncertainty may underestimate the true uncertainty.

- The following are more minor comments, but taken together they make the overall model description difficult to follow, which is a major concern:

Response Thank you for highlighting these points. Please consider our point-by-point response below.

- line 846: “Re-estimate sparse model”. This is the first time a sparse model is mentioned in the paper. From looking at the cited modeling references, I believe this means something more like “fit the model with the selected covariates”.

Response Thank you for pointing out this lack of clarity and we agree with this comment. To make this point clear, we explain this in more detail as follows:

Re-estimating the model omitting non-informative covariates

The final model – i.e., the model containing only those covariates in each predictor η_k that were selected in the previous step – is estimated using classical MCMC sampling techniques.

- Line 891: “to allow the ST effects [missing word: to] vary smoothly with the sex of the child...” The meaning of this is unclear to me. Are the outputs somehow smoothing over the binary sex variable? Or is this trying to convey that the ST effects vary smoothly conditional on sex? If so, what about the other categorical covariates?

Response We apologize for the ambiguity. To stratify our estimates by sex – and thus individual risk maps for boys and girls – a varying coefficient term using the sex of the child as the effect modifier is included for the complex space-time and space-age interactions. Thus, the corresponding surfaces are modified by the sex of the child in an unspecified smooth way. We have revised our manuscript for clarity as follows:

In addition, to stratify the results by sex the spatio-temporal effects vary smoothly with the sex of the child, i.e., the complex space-time effects include varying coefficient terms [10], where the sex of the child is used as an effect modifier of the smooth space-time interaction.

Hypothetically, similar interactions are possible with other covariates, not just categorical ones. However, this leads to an extreme increase in computation time without – in our opinion – adding much value.

- With the described multi-step process, is it possible (or better yet straightforward) for someone to reproduce these results?

Response Thank you for pointing out this important issue. We also provide an example (see R-script **example.R**) that illustrates all steps of the estimation routine outlined in Section *S 3 Methodology* of the Supplementary Materials, which briefly illustrates the individual computational steps using an example based on simulated data. In addition, all the relevant R-scripts are provided in the zip-file *R-scripts_Anemia-paper.zip*. Note that the analysis uses custom tailored code for the HPC infrastructure we had access to, written in R [11] using the packages mentioned in this Section *S 4 Software and Computational Details*, and we will provide assistance with these scripts upon reasonable request. In addition, note that we do not have permission by DHS to pass on the DHS survey data to other researchers.

- line 940: “Non-informative covariates are selected... using the algorithm allows [missing word: one/the model] to select only ... terms ... with the greatest contribution ...” Are the non-informative covariates removed or selected?

Response Thank you for pointing out this lack of clarity. In this step (i.e., defining candidate predictors), the non-informative covariates are identified using the method described in detail in [9]. Then, in the next step (i.e., re-estimating the model omitting non-informative covariates), the model is re-estimated using MCMC sampling techniques, including only those covariates in each predictor η_k that were selected in the previous step. Thus non-informative covariates of each predictor η_k are omitted from the final model. To make this clearer, we have reworded these sentences as follows:

Non-informative covariates are detected by the application of a novel backfitting algorithm described in [9]. Using this algorithm, only those model terms in each predictor η_k that make the largest contribution to the individual out-of-sample log-likelihood of each predictor η_k are selected.

4.2 Anemia

1. Was any attempt made to use the survey DHS provided survey weights? If yes, how? If not, what are the implications of ignoring them?

Response Thank you very much for pointing this out and we apologize for any confusion that may have resulted. DHS survey weights were used for all descriptive results and to calculate the country level prevalence. In addition, as noted in [12], they find no difference in the overall mortality rate predicted by the weighted and unweighted models when considering under-five mortality. Furthermore, they argue that this is also true for a more general class of regression models. Thus, in this context, the survey weights seem to be important only when calculating descriptive statistics from the data and when aggregating the estimates to, for example, the country level.

4.3 Results

1. One of the great advantages of a full probabilistic model is that it quantifies the uncertainty of the generated estimates. I don't believe I saw any results which showed or discussed the uncertainty present in these results. I encourage the authors to provide some visualizations showing (mapping) the uncertainty of the anemia prevalence estimates since the magnitude of this greatly impacts the interpretation of the results. Without understanding the uncertainty of the results, it makes it impossible to assess, for example, if the non-linear impact of temperature on prevalence in India and Madagascar really should be interpreted as a positive association (as described in lines 172-174) or if it is not statistically discernible from a flat line.

Response Thank you for pointing this out. We have now added *Figure 2*, which shows the prevalence of severe, moderate, and mild anemia for selected covariates and locations, along with 95% credible intervals of the estimated effects. Similarly, in *Figure 4*, which shows the predicted spatio-temporal prevalence of any form of anemia along with 95% credible intervals. In addition, *Table S 6.2 to S 6.5* show the prevalence of any, mild, moderate, and severe anemia at the country level along with the corresponding WHO estimates [13, 14] and their respective 95% credible intervals.

2. The results paragraph (lines 212-221) seems quite general and vague. All this wonderful effort has gone into the analysis and a few more specific examples of insights gleaned seems warranted. I believe a more detailed results section would more clearly demonstrate the benefits of this method and what this paper adds to the scientific literature.

Response Thank you for the comment that we support wholeheartedly. In the revised version of our manuscript, we have extended the results section providing more concrete examples. We also included a more in-depth comparison of our findings with results from the previous literature in the discussion section.

3. Somewhat related, the authors allude, multiple times (eg line 58,), to their model's ability to provide "insights into the underlying causes of anemia," but I didn't find much discussion of these, especially given the number of included covariates.

Response The reviewer is right. We have carefully revised this part of the introduction to focus attention on our main goal – to provide a much-needed spatiotemporal high-resolution mapping of anemia. The rest of the introduction is now structured as follows:

In addition, this framework allows us to determine the association of anemia with climatic, environmental, and socio-economic factors using a novel data set for 37 LMICs. In doing so, we will provide valuable insights into the spatio-temporal distribution of this important global public health determinant and its association with its correlates.

5 Minor Remarks

1. Table S1: under 'Conflict' it states "1 if more than 5 conflict incidents ... ", yet in Tables S3 and S4 it states "Indicator whether a conflict...". Is the variable 1 for one or more, or for 5 or more conflicts?

Response Thank you for noting this inaccuracy. The conflict variable takes the value of one if more than five conflicts have been observed in the past within the specified buffer around the primary sampling unit. In the manuscript, we have now corrected and revised this.

2. Table S4: model term f_{22} is described as spatio-temporal but it has no time index. Perhaps you meant spatial-age effect?

Response The reviewer is correct. Spatio-temporal is not appropriate in this context, and as the reviewer points out, the spatial-age effect is more appropriate. The model term f_{22} corresponds to the three-dimensional tensor product spline between the longitude and latitude coordinates of the primary sampling unit and the age of the child (in months). This allows for an age-specific spatial effect.

3. Lines 131-136: there are some grammatical errors which make this sentence hard to parse. In addition, this is the first time the “naive models” are mentioned, making it difficult/impossible for the reader to correctly interpret these validation metrics.

Response We agree with the reviewer that this sentence was hard to parse. Hence, we corrected this sentence, paying extra care to improve the readability of this sentence. In addition, with “naive model” we refer to the model that relates the Hb level of each subregion to the normal distribution including only the intercept of the two distributional parameters μ and σ . Please note that to improve readability this part was moved to the Section *S 5.1 Calibration and convergence checks* of the Supplementary Materials. Now the sentence is structured as follows:

Comparing the MSE of each final subregional model, calculated on the test data – i.e., data that were not used for modeling – with the subregional “naive model” – i.e., modeling the Hb level for each subregion based on the parameters of the normal distribution omitting all covariates – using the skill score [15], the improvement ranges from 16.3% to 19.7%, highlighting a substantial improvement in the predictive ability of the final models compared to these “naive models”. Similarly, using the same metric for the CRPS [16] the improvement ranges from 9.6% to 11.9%, highlighting this improvement.

4. I’ve already pointed out a few minor corrections in the above quotes. Here’s a few others I noticed:

Response We appreciate your feedback and have corrected the aspects you highlighted, and hope that we have addressed your suggestions.

- Line 136: Similar → Similarly.

Response Thank you for pointing out these grammatical inconsistencies. We have corrected these errors and thoroughly proofread the manuscript at multiple venues prior to resubmission.

- Line 900: missing section number. (I think I saw another missing section number somewhere in the appendix, but I didn’t note where.)

Response The reviewer is correct we did not include section numbers in the Supplementary Materials. We have now followed the style and formatting guide of **Communications Medicine** to ensure that our manuscript conforms to the chosen formatting standard. The revised manuscript now includes section numbering in the Supplementary Materials.

- The title to Table S3 is hard to parse with multiple “included”s and “model”s.

Response We have reworded the heading of *Table S 3*, which now reads as follows:

Covariates in each predictor η_k included in the full model of the selection stage used for the estimation of the distribution of \mathbf{Hb} .

It is our sincere hope that we have addressed all of your remaining concerns, and our apologies once again for any oversights on our part. We greatly appreciate all of your feedback, and we are confident that the manuscript has been significantly improved as a result of it. Thank you again for your valuable input.

Comments Reviewer # 2

General comments

Congratulations to the team, a very well built and written paper. It provides interesting insights on anaemia. The steps followed to understand the data, distributions, selecting covariates and model building to validation is commendable. The analytical methods are well elaborated.

Minor comments that the authors need to consider to make some sections clearer, revisit the texts for a more focused narratives and correct few mistakes.

Response We thank Reviewer # 2 for the overall positive feedback. We addressed all comments and questions point by point below and sincerely hope that we have cleared up any remaining ambiguities.

Specific comments

- Abstract: I suggest to include some of the main insights from the work in the abstract if possible – at the moment it weights too much on data and methods.

Response Thank you for your suggestion. We have revised the abstract accordingly, taking special care to follow the formatting guidelines of **Communications Medicine**. The corresponding paragraph *Results* now reads as follows:

This analysis provides high-resolution estimates for all forms of anemia and reveals and identifies striking disparities within and between countries. Based on these estimates, the prevalence of anemia decreased from 65.0% [62.6% – 67.4%] in sub-Saharan Africa and 63.1% [60.6% – 65.5%] in 2010 to 63.4% [60.7% – 66.0%] in sub-Saharan Africa and 58.8% [56.4% – 61.3%] in South Asia. This translates into approximately 98.7 million and 95.1 million affected children aged 6 to 59 months in 2020, respectively, making it a major public health concern.

- In all these sections – distribution of Hb, Probabilistic distributional regression, Variable selection, Calibration and convergence checks, etc – some details could be omitted to make the narratives precise and succinct. Lengthy descriptions make the paper too long unnecessarily. E.g., stating training and test data should be simplified as that doesn't need too much elaboration or could just add a reference for interested readers.

Response Thank you for pointing this out. We have substantially shortened the mentioned sections and moved all the details to the Supplementary Materials when we thought that some aspects should be elaborated in more detail. We paid special attention to precise and clear writing to condense these sections of the manuscript.

- Understanding the regional disconnectedness, and the approach used by the author to model the regions separately, however, that may bring a query on input data – to clear this, I would suggest including in the Supp materials maps indicating locations of the observed PSU for SSA, Madagascar and South Asia.

Response Thank you very much for this valuable feedback. We have added a map showing the primary sampling units and the included countries (Figure S 2) to the Supplementary Materials.

- Lines 172-176: The statement “This finding is particularly concerning, as rising temperatures due to climate change could potentially exacerbate the prevalence of anemia in sub-Saharan Africa” seems like an over emphasis on the effects of climate on Anemia in SSA.

Thank you for pointing out this ambiguity and unclear wording. We have carefully revised the entire paragraph to make it clearer and consistent with our main message here that it is important to identify potential pathways through which rising temperatures may lead to increases in anemia prevalence, particularly in South Asia. The revised paragraph now reads as follows:

Examining the bottom panel of *Figure 2*, which shows the estimated effect of surface temperature, along with 95% credible intervals on the prevalence of any form of anemia for selected locations. A relatively constant effect is observed in mainland sub-Saharan Africa and Madagascar. In contrast, in South Asia, there is a shallow but steadily increasing association with increasing surface temperature, suggesting that within South Asia, the prevalence of any form of anemia increases with increasing surface temperature. Furthermore, the estimated credible intervals within South Asia are narrower than the credible intervals in sub-Saharan Africa, highlighting the importance of this effect within South Asia. Accordingly, in areas expected to be severely affected by climate change, this relationship and potential pathways by which rising temperatures lead to an increase in anemia prevalence need to be closely monitored to better understand the impact of rising temperatures and their mechanisms on anemia prevalence.

- What was the basis and criteria for presenting results of selected age categories Eg in Figure 4? Why not blocks of age say 6– 18m, 18-30m, etc? It is a bit hard to use a-month-specific results for practical recommendations, in my opinion.

Response Thank you for pointing this out. The reason for the month-specific estimates is twofold. First, to our knowledge, this is the first study to estimate the prevalence with this precision. Second, we find a high degree of non-linearity in the association between Hb levels and the age of the children. Thus, binning of age may introduce imprecision. However, we agree with Reviewer # 2 that the binning age is more meaningful for practical purposes. In the paper, we now show the results for the following age bins: (6, 24]; (24, 42]; (42; 60]. The monthly estimates are available from us upon reasonable request.

- Lines 200 onwards: Fair to weigh the descriptions a lot on the problematic areas, increasing trends observed, etc, but would be also great to speak about the reduction observed in some countries both within and outside SSA. This is done but I think on a very minimal level.

Response Thank you for the comment and we agree with the reviewer. In the revised version of our manuscript, we have extended the results section providing more concrete examples of countries and regions that experienced increases as well as decreases in the prevalence of anemia over time. We also included a more in-depth comparison of our findings with results from the previous literature in the discussion section.

- I would present results of both boys and girls as main figures in the manuscript or as Supp material. Separating them creates ambiguities to comprehend what is going on. If the figures are too many you could opt to only present specific categories.

Response Thank you for pointing out this ambiguity. We have corrected this in the main manuscript, where we present the results for anemia without stratifying by sex. The results for both boys and girls are now shown in the Supplementary Materials.

- The subsection on “Parasites”: The section described important variables for existence of Hookworm larvae, but not clearly stating what variables were used as a proxy for it in the analysis. Kindly check and ensure clarity.

Response Thank you for pointing out this ambiguity. We have revised this section accordingly and have now include literature on which covariates can be considered appropriate proxies for these parasites and which covariates we have included. The paragraph has been completely revised to read as follows:

Parasites such as hookworm or schistosomes are considered to be causative agents of anemia [17–19]. There are several distinct remotely sensed covariates that have been found to be associated with these parasites. For example, Brooker and colleagues [20] found that hookworm prevalence is correlated with surface temperature, altitude, and the normalized difference vegetation index (NDVI). Karigiannis-Voules et al. [21] highlighted a positive association between the prevalence of soil-transmitted helminths (e.g., hookworm, *Ascaris lumbricoides*, and *Trichuris trichiura*) and surface temperature and precipitation. In addition, hookworm larvae require specific temperature, soil, and aridity conditions to be able to hatch [22].

For example, Kokaliaris et al. [23] find the risk of schistosomiasis prevalence among children in sub-Saharan Africa to be linked to precipitation, NDVI, and the distance to the nearest body of fresh water. Similarly, Lai et al. [24], for example, highlight the importance of land cover classification, distance to the nearest body of water, or surface temperature on the risk of different *Schistosoma* species.

Thus elevation, distance to the nearest body of water, land cover, precipitation, NDVI, and 2 m surface temperature are used as proxies for the prevalence of hookworm and schistosomes, see *Table S 1* for the source of these covariates and *Table S 3* for details on how these covariates were included.

- One of the common analytical approaches to model anemia data by these classes is a multinomial (logistic) model. Authors described (pros/cons) various approaches but not this one – I am curious to know why?

Response Thanks for bringing this up. There was no specific reason not to mention multinomial (logistic) regression. During our literature review, we put an emphasis on studies that focused on spatial prediction of anemia in the study area. We did not identify any study that used (ordered) multinomial regression models. To make the picture more complete we revisited the literature and identified two articles for single countries using approaches similar to multinomial regression [25, 26].

In our analysis of anemia, we chose to relate the Hb level to a continuous distribution for three main reasons: First, anemia can be thought of as an ordered response (i.e., severe anemia < moderate anemia < mild anemia < no anemia); using a multinomial model would be a model misspecification. To overcome this remedy, an ordered multinomial model would be more appropriate, but the underlying proportional odds assumption may be too restrictive. Second, discretizing the Hb measurements to determine anemia status potentially discards a lot of information in the discretization process. Third, different cut-offs for anemia have been reported for anemia in the literature [27]. By using an approach based on modeling a continuous distribution, it is easy to calculate e.g., prevalences based on different specified cut-offs. In contrast, for all approaches that discretize the Hb level, the model would have to be re-estimated altering the specified cut-offs.

Editorials

Figure 2:

- As this is among your main Figures for the manuscript, the caption should include explanation of the abbreviations/ISO3 used – this may not be trivial to all readers.

Response Thanks for pointing this out. We have added an explanation of the ISO3 country code abbreviations (i.e., ETH stands for Ethiopia, BWA for Botswana, IND for India, and MDG for Madagascar) used in the corresponding caption.

- Line 161 – with the observed trends I wouldn't say this is highly negatively correlated, could the authors elaborate more here? What measure has been used to come with that quantifiable conclusion? The trends observed by age are steeper than the ones for HHD wealth – surprisingly this was not mentioned.

Response We agree with Reviewer # 2 that in the initially submitted manuscript too much emphasis was placed on household wealth and not enough on the age-specific effect. In the revised version, we put more emphasis on the nonlinearity of the age-specific effect, which also underscores the importance of stratifying the estimates of any health outcome by age. Otherwise, this may lead to model misspecification. The paragraph on household wealth was modified as follows:

Anemia and household wealth are negatively correlated, although the effect is less pronounced than the age effect, with the prevalence being highest among children living in poor households. The correlation between anemia and household wealth suggests that food scarcity and the lack of nutritious staple foods may contribute significantly to adverse health outcomes among members of disadvantaged households. This association underscores the importance of addressing socio-economic factors and improving access to nutritious food to reduce the negative impact of anemia and other health-related outcomes.

Footnote 5:

- Not sure I understood this part – as at birth and directly after birth the Hb levels are usually higher and distort possibly the estimates. Could the authors elaborate this further/ make it clear.

Response Routinely the DHS only collect Hb measurements for children older than 5 months. This is because at birth and directly after birth, the Hb levels are usually higher (possibly due to better in-utero provision of nutrients) and accordingly can distort the estimation of the prevalence of anemia. Moreover, the WHO refers to the same age group for a similar reasoning. We have slightly reworded this to enhance clarity, as follows:

Note that the DHS does not provide information on the Hb levels in children under 6 months of age. This is because Hb levels are usually higher at birth and within a few months after birth, possibly due to better provision with nutrients in-utero, which may bias the estimated prevalence of anemia [28].

Fig. S1

- The red/green dots are not really visible. Check for alternative colours.

Response Our apologies for this unnecessarily poor choice of color combinations, and the fix is in place.

From line 177 – Spatio-temporal dynamics

- Authors should revisit this section and check that the narratives are correct, referencing of Figures is not confusing, and all the captions are correct – for example Fig 4 is referred to as for girls in line 183 and as for young children in line 197; the caption for Fig 4 mentioned the results are for 12 months but the figure shows for 6months, etc.

Response We apologize that some references and labels got mixed up. We have carefully checked that all labels and captions now correspond to the correct Figure and are referenced correctly in the manuscript. Please note that we now provide the spatio-temporal risk maps without stratification by sex in the main text and the stratified risk maps in the Supplementary Materials. This helps to provide a better overall picture. Please also note that in the revised manuscript we now provide the estimates aggregated for 18 month age bins. This does not alter the overall picture. The monthly estimates are available from the authors upon request.

We apologize again for any oversights on our part, and we sincerely hope that we have addressed any remaining concerns. Your insightful feedback was very much appreciated, and we are confident that the manuscript is a marked improvement as a result. Thank you for your valuable input.

Comments Reviewer # 3

Summary

The manuscript “High-resolution spatial prediction of anemia risk among children between 6 and 59 months in low and middle-income countries” uses a fully Bayesian probabilistic distributional regression modelling framework to estimate the prevalence of mild, moderate, and severe anaemia in children under five years of age in sub-Saharan Africa and South Asia. The model used anaemia measurements from DHS surveys and a selection of spatial-temporal covariates such as temperature, land coverage, and malaria prevalence to predict the prevalence of anaemia at a 20x20km spatial resolution.

Estimates of the prevalence of mild, moderate, severe and any anaemia are presented for children at different ages for 2010, 2015 and 2020. They highlight anaemia being highest in young children (6 months) and show little temporal changes in burden. The highest levels of anaemia were identified in western sub-Saharan Africa and parts of east Africa. This presents the first subnational estimates of childhood anaemia in these regions and may prove a useful tool for benchmarking and decision making to tackle anaemia and food insecurities (SDG2) in children.

General comments to author

Overall, I found this an interesting, well conducted study. The modelling strategy appears well reasoned and appropriate to the data and the figures presenting the results are clear and informative. However, I think that it could benefit from a more structured methods section and a more quantitative assessment of the results. Specifically:

Response We thank Reviewer # 3 for the overall positive and constructive feedback. Below is our point-by-point response to the open questions, comments, and ambiguities raised during the review.

1. I found myself having to go to the supplementary materials to follow the methods. I think that it would be beneficial to have a brief overview of the input data and the modelling framework at the beginning of the methods section and then go into brief details of the probabilistic regression model, distribution choice, variable selection, model calibration/convergence. Whilst I assumed the DHS anaemia measurements were used as input data I had to go to the supplement to confirm this. More balance between the main text and supplementary methods is required.

Response We apologize for the sub-optimal organization of the manuscript. In the revised manuscript, we have focused on the points raised by the three reviewers. Hence, we have significantly shortened the methods section, while still providing the necessary information to be able to follow our modeling approach. Thus, the manuscript provides a concise overview of the five modeling steps. Details of these steps, as well as the performed pre- and postprocessing steps, are provided in Sections *S 1 Data* to *S 3 Methodology* of the Supplementary Materials.

2. I think that the results could be strengthened and made more impactful by including the a summary of the estimates of the prevalence of anaemia from your model. For example, I would like to see the overall prevalence and total number of children estimated to have mild/moderate/severe anaemia in 2010 and 2020 in your study area (mean and uncertainty intervals) as well as the values for countries or locations you identify as hot spots. This would help to put the results into context and enable comparisons to previous estimates.

Response Thank you for highlighting this. We now present the estimated prevalence of any, mild, moderate, and severe anemia (including 95% credible intervals), for the years 2010, 2019, and 2020 in *Table S 6.2* to *Table S 6.5* in the Supplementary Materials. These tables also include the corresponding estimates of Stevens et al. [13] / the WHO [14] for the years 2010 and 2019 (the most recent year).

In addition, these tables include estimates of the population at risk that are based on the UN World Population Prospects [29]. Thus, the total number of children with anemia can be calculated easily based on our estimates and those put forth by WHO.

3. One of the strengths of this study is it being a subnational analysis. It would be interesting to assess the in-country variation in anaemia prevalence to highlight inequalities in determinants of anaemia such as food security and infectious disease prevalence. Using metrics such as the Gini coefficient or calculating the mean relative deviation are potential methods which I would like to see explored. Additionally, it would be interesting to analyse the data by urban/rural classification to look at the disparities there.

Response Thanks for highlighting these two aspects. We completely agree with the reviewer that our analysis also allows to study inequalities within countries. Hence, we added *Figure 3* to the main part of the manuscript that highlights the within- and between-country inequality in the prevalence of anemia. Due to space constraints, we decided against adding other metrics of inequality but we take this point seriously and explore this in more detail as a follow-up research project. For the latter point, our estimates are implicitly stratified by urban/rural classification by including the land cover classification as covariate. This covariate includes, in addition to different vegetation classes (e.g., *Forests, Shrublands, Savannas, Wetlands, ...*) also the class *Urban and Built-up Lands* ensuring that the estimation results are stratified by this classification. Note that this variable has been selected in the regional sub-model for South Asia and Madagascar.

4. Did you find data sparsity and issue as you are estimating for many countries with no data. Did you test the model holding out whole countries worth of data to assess the model performance here?

Response To strengthen our estimates, we did a cross-validation procedure, where in each model run, for each country individually, observations from country i were removed from the training data, and the models were then validated on the validation data including the observations from country i . Please note that since only four countries (i.e., Bangladesh, India, Myanmar, and Nepal) were included in the regional model for South Asia, instead of leaving out India entirely, the cross-validation was done by leaving out admin-1 regions within India and the three consecutive countries. Note also that for Madagascar, the cross-validation was done by omitting admin-1 regions individually.

During this evaluation process, we observed stable estimates across all runs, and the diagnostic checks – e.g., PIT-histogram, worm plot, QQ-plot, all evaluated at the out-of-sample data – looked similar for each run and did not indicate any problems caused by data sparseness. To shed more light on this, we also calculated the relative deviation of e.g., the CRPS to the model without withholding out any observations from the training data. For details see Section 5.3 *Results potential data sparseness issues – leave one administrative region out cross-validation* of the Supplementary Materials.

5. In the discussion I think that it is important to compare you're results to previous estimates of anaemia burden. How so these compare to the likes Stevens et al. [13] (Ref 34 in this manuscript).

Response Thank you for the comment and we agree with the reviewer. In the revised version of our manuscript, we have rewritten and extended the discussion section and now provide a more in-depth comparison of our findings with results from the previous literature.

Specific Comments to author:

Introduction

- Lines 37-40: You state that studies on the determinants of anaemia and therefore knowledge of the trends and burden across LMICs is limited. I think here it is worth noting the estimates from Stevens et al. [13] (Ref 34 in this manuscript) as they provide national estimates of anaemia burden in children u5 for 2000-2019, giving you another estimate of anaemia across LMICs.

Response Thank you for pointing out this shortcoming and guiding us in this direction. In the introduction of the previously submitted manuscript, we focused on studies providing estimates of the burden of anemia at the subnational level, omitting the WHO country estimates that are based on the aforementioned article by Stevens [13]. To complete the picture and to draw attention to this important first attempt, we have restructured this section as follows:

However, when it comes to childhood anemia, studies that provide high-resolution estimates of the burden of anemia and that analyze potential determinants of anemia are limited to single countries [25, 30–32] or small groups of countries [33, 34]. Estimates of levels and trends across geopolitical regions are limited to the study by Stevens and colleagues [13], which provides an almost global assessment of anemia risk at the country level and is shared by the WHO [14]. Thus, the knowledge of the levels and trends beyond country estimates of anemia in LMICs is still limited to individual countries, or small groups of countries.

- Line 57-60: you list variables your study will provide insights into for their contribution to the causes of anaemia, however, these aren't included as covariates in the model so this is an inaccurate statement.

Response Thank you for bringing this ambiguity to our attention. We have revised this section and put more emphasis on our main goal – the analysis of spatio-temporal dynamics. However, a major advantage of using distributional regression is that the estimated effects of covariates remain interpretable. To illustrate this, *Figure 2* shows the estimated prevalence of severe, moderate, and mild anemia as a function of selected covariates for selected locations. In addition, *Table S 3* included in the Supplementary Materials lists all included climatic, environmental, and socio-economic covariates.

Methods

Measurement of anemia

- Line 68-69: To be pedantic, the definition of anaemia is $< 110\text{g/L}$ therefore non anaemic would be greater than or equal to 110g/L (could reword to make this consistent with other locations in your manuscript).

Response Thanks for spotting this inaccuracy. We changed the wording to be consistent with the remainder of the manuscript.

- Need to link this section to the DHS surveys and clarify this is where you are getting your measurements of anaemia from. Are you using the adjusted or unadjusted Hg levels from DHS?

Response Thanks for spotting the missing information. We added a footnote (footnote 1) to the manuscript, where we mention the most common methods to measure the Hb level and also the DHS surveys as our data source of Hb measurements. Further details on the data are given in the Supplementary Materials.

For modeling we use the adjusted Hb levels, as the distinction between anemic and non-anemic is based on these values.

Distribution of Hb levels

- Lines 73-93: An overview of the modelling strategy before going into the details of the data distribution would be beneficial and put this in context. I think that only a brief description of the Hb level distribution and the importance of selecting an appropriate one (a couple of sentences) is required here and most of this section would be better suited to the supplement.

Response We have completely revised this section of the main part of the manuscript and condensed it considerably. Now, only the five key aspects of the analytical framework are highlighted and briefly described. For interested readers, we provide a thorough elaboration of the analytical framework in Section *S 3 Methodology* of the Supplementary Materials.

Probabilistic distributional regression

- Line 142-143: the correlation between input data and prediction at the country level is very good. But as this is a subnational model, how do the subnational comparisons look (e.g., at administrative division 1)?

Response In addition, we have added the scatterplot between the reported prevalence (calculated from the DHS surveys) at admin-0 (*Figure S 14*) and admin-2 (*Figure S 13*) and our estimates to the Supplementary Materials of the manuscript. See *Figure 1* for the scatterplot of the predicted prevalence of any form of anemia against the reported prevalence at the admin-2 level. See also *Figure 2* that shows the estimated Hb level against the observed Hb level aggregated to admin-2 regions. This is also included in the Supplementary Materials (*Figure S 12*). Overall, the correlation between the prevalence calculated from DHS and our estimates is also very high at these sub-national levels. Our model seems to perform worst for mild anemia. However, this is not, surprising given the narrow cut-offs (i.e., $P(100 \text{ g L}^{-1} \leq \text{Hb} < 110 \text{ g L}^{-1})$) used to define mild anemia.

- I would like to see a bit more of a description of the model here, including what package and software were used to fit it. Please include a location where the code used to fit the model is shared.

Response We now include the Section *S 4 Software and Computational Details* in the Supplementary Materials. This section includes information on the used R-packages and the R-script `example.R` that illustrates all steps of the estimation routine using an example based on simulated data. In addition, all the relevant R-scripts are provided in the zip-file *R-scripts_Anemia-paper.zip*. Note that the analysis uses custom tailored code for the HPC infrastructure we had access to, written in R [11] using the packages mentioned in this section. In addition, note that we do not have permission by DHS to pass on the DHS survey data to other researchers. We also note that further details can be received upon reasonable request.

Results

Effect of covariates (marginal probabilities)

- Line 173: Clarify in the text here that the “positive association” you are referring to is with temperature. I think discussion of the strength of this association is also important as it seems quite a weak association in India.

Response Reviewer # 3 is correct with this statement that this formulation was somewhat misleading. We have revised this paragraph to emphasize our observation that this association needs to be closely monitored, particularly in South Asia, and that it is important to identify potential pathways through, which rising temperatures may can lead to an increase in anemia prevalence. The revised paragraph now reads as follows:

Examining the bottom panel of *Figure 2*, which shows the estimated effect of surface temperature, along with 95% credible intervals on the prevalence of any form of anemia for selected locations. A relatively constant effect is observed in mainland sub-Saharan Africa and Madagascar. In contrast, in South Asia, there is a shallow but steadily increasing association with increasing surface temperature, suggesting that within South Asia, the prevalence of any form of anemia increases with increasing surface temperature. Furthermore, the estimated credible intervals within South Asia are narrower

than the credible intervals in sub-Saharan Africa, highlighting the importance of this effect within South Asia. Accordingly, in areas expected to be severely affected by climate change, this relationship and potential pathways by which rising temperatures lead to an increase in anemia prevalence need to be closely monitored to better understand the impact of rising temperatures and their mechanisms on anemia prevalence.

- Line 174-6: The conclusion that climate change is going to potentially exacerbate anaemia in sSA quite a leap from these results, when the issue is far more complex with temperature/rain fall/crop yields impacting food security and other confounding factors. I think this is something more suited to being expanded on in the discussion, with the inclusion of other referenced evidence.

Response Reviewer # 3 is correct in this observation, which points in the same direction as the previous comment. We have carefully revised this section accordingly. The point of the statement is simply to emphasize that, from our perspective, it is important to closely monitor this association, particularly in South Asia, and to indicate that rising temperatures may lead to an increase in anemia prevalence. The entire paragraph was revised as follows:

Examining the bottom panel of *Figure 2*, which shows the estimated effect of surface temperature, along with 95% credible intervals on the prevalence of any form of anemia for selected locations. A relatively constant effect is observed in mainland sub-Saharan Africa and Madagascar. In contrast, in South Asia, there is a shallow but steadily increasing association with increasing surface temperature, suggesting that within South Asia, the prevalence of any form of anemia increases with increasing surface temperature. Furthermore, the estimated credible intervals within South Asia are narrower than the credible intervals in sub-Saharan Africa, highlighting the importance of this effect within South Asia. Accordingly, in areas expected to be severely affected by climate change, this relationship and potential pathways by which rising temperatures might lead to an increase in anemia prevalence warrants close monitoring to better understand the impact of rising temperatures and their mechanisms on anemia prevalence.

Spatio-temporal dynamics

- Line 185-8: This is a very interesting result. Why may this be the case? Are other determinants of anaemia increasing if malnutrition is decreasing (malaria burden also decreases in this time frame but what about schistosomiasis etc)? I think this warrants further attention in the discussion section.

Response Reviewer # 3 is right about that. We tried to find a systematic reason why this might be the case, but are still unclear. This is also supported by the fact that anemia is multifactorial [13, 35] and therefore it is difficult to determine exactly why this might be the case.

We have also tried to take this into account in the manuscript by including the following outlook.

This suggests that anemia may be the result of a complex and multifactorial disease history [13, 35]. It also indicates that there may be a correlation between populations with a high prevalence of anemia and those with a high prevalence of the diseases that cause anemia. Furthermore, it highlights that this population group may face certain disadvantages.

- Line 190: “Many countries showed a(n) increase in the prevalence level of anaemia” – I would like to see some detail on this, how many countries saw an increase, which countries increased the most, what was the level of increase. Corresponding numbers from the modelled estimates to back up this assertion would strengthen the results.

Response Thank you for this comment. In the revised version of the manuscript we now provide a much more detailed description of countries that have experienced an increase/decrease in the prevalence of anaemia over time.

- Line 191-193: I don't think this statement of a global comparison is required here, I would prefer more information on your results here and to compare this to other estimates in the discussion.

Response Thank you for the comments. We deleted this statement and now provide a more detailed comparison of our findings to the existing literature in the discussion section.

- Line 194: 'Substantial burden' – what was the burden? How many cases of anaemia in 2020 does your model estimate?

Response Reviewer # 3 is correct that a more detailed presentation of the results would be particularly beneficial to practitioners. We added more content on this in a separate paragraph to the Section *Results* and used the UN World Population Prospects [29] to quantify the estimated prevalence of anemia, which is about 194 million children in sub-Saharan Africa (98.7 million affected children) and South Asia (95.1 million affected children) in 2020. This is very close to the combined estimate (i.e., 192 million children) by Stevens and colleagues [13] for 2019 (the latest year for which they provide estimates on anemia prevalence).

- Line 196: states that you have identified hot spots. You repeat this assertion with a bit more info in lines 202-206 – not needed in both places. I would like to see some numbers to accompany your observation of 'hotspots'. How high does the prevalence of anaemia reach in these locations? It would also be interesting to note and compare this to cold spots.

Response Thank you for pointing this out. We have completely revised the Section *Results* to include more detailed information on hotspots and coldspots. In addition, to strengthen our results, we have included model-based estimates for several admin-1 regions that are consistent with the included high-resolution maps. To further strengthen our estimates, we compare these estimates at the country level with WHO estimates [13, 14].

- Line 197-198: Again, having some numbers from your model would be good to quantify this statement.

Response Please see our response to your previous comment. We added numbers to strengthen and quantify our observations.

- Line 214-216: Again, some numbers would be good here.

Response Please see our response to your previous comment. We added numbers to strengthen and quantify our observations.

- Fig 3: Could the same colour ramp be used as in figs 4 & 5?

Response We have changed the color palette of *Figure 4* to match the color palette used in *Figure 4* and *Figure 5*. In addition, please note that we now present results in the main text without stratification by sex, as suggested by Reviewer # 2. More specific results on the prevalence of anemia stratified by sex are now provided in the Supplementary Materials.

- Fig 4 & 5: Visually there appears to be little difference between the maps for girls and boys. This is the first mention of the analysis being stratified by sex and it isn't discussed further. If there are no noteworthy differences between the boys and girls to be discussed could these figures be shown for the sexes combined (and then specify differences in the supplement)?

Response Even though the disparities between boys and girls appear to be negligible, the sex of the children got selected in the regional sub models for sub-Saharan Africa and Madagascar. To improve readability we only show the results in the main text without stratification by sex and discuss this briefly. More detailed results on the prevalence of anemia for boys and girls are now provided in the Supplementary Materials *Figure S 17* to *Figure S 22*.

- Fig 4: It looks like there is a decrease in anaemia in western sSA for the 59 months map in 2020. This is contrary to what you have said in the text, is this an important result?

Response We carefully checked the text to match what is depicted in the Figures. The reviewer is correct that overall anemia slightly decreased for the corresponding age in western sub-Saharan Africa. We now discuss this positive development in more detail in the main part of the manuscript. Note that we now provide the estimates aggregated for 18 months bins in the revised manuscript. However, the overall picture does not change. The monthly estimates are available from the authors upon request.

Discussion

- Line 230-233: Yes, the predictive performance of the model at country level is impressive but as what is novel about this study is the subnational estimates it would be interesting to assess the performance sub-nationally too. How did you select your held out data, was this random or stratified over time and location? Unless whole countries or surveys have been held out I would expect the country level correlation to be high as there is a wealth of other point level data from that country-year included in the model. Can you describe this model validation a bit further (not required in main text but in the supplement this could provide strength to your assertion that your model has excellent predictive performance)

Response Thank you for pointing this out. This was also raised by the two other reviewers and we want to respond to each of the concerns raised individually, but there may be some overlap in our response. We have added two new paragraphs to the Section *S 3 Methodology* of the Supplementary Materials, namely the Paragraph *Creating the training and test data sets* and the Paragraph *Robustness check for potential data sparseness issues – leave one administrative region out cross-validation*, where we describe in detail the training and test data set generation and the additional validation procedure. A concise summary of these two paragraphs is the following answer, which is identical to the response to a point raised by Reviewer# 1:

Thank you for raising this clarification issue. We are aware that random sampling can be problematic in this case. As now shown in the flowchart included in the Supplementary Materials as *Figure S 1*, we accounted for this by following the procedure described by Gething and colleagues [8], which is based on Delaunay triangulation, where we used $\sqrt{\text{area}_i}$ as sampling probability. In fact, to explore the consequences of neglecting this important aspect, we also ran the models with the training and validation data split randomly. Between the two methods, the results are quite similar – due to the large number of observations. In addition, to validate this approach, we validated our results using cross-validation, where we ran the final model by country, and for each run, the observations of the training data pertaining to country i were held out. Then, the test data – which include observations from the omitted country – were used to assess the effect of omitting country i on the estimates.

- Line 240-242: “This choice reflects the fact that most fatal cases and infections are caused by the dominant species *P. falciparum*, which is prevalent in sub-Saharan Africa.” I don’t think this is an accurate assessment of what the inclusion of malaria incidence in the model reflects, especially as you are not looking at fatalities here. Rather it highlights the importance of Pf malaria as an aetiology of anaemia in these regions. Anaemia due to Pf will likely be due to either repeat infections or untreated infections progressing to severe malaria. Evidence suggests that in high Pf locations severe malaria presents in children at a young age as severe malarial anaemia, which may be reflected in your results. I think this in an important point that warrants discussion.

Response Thank you for pointing out this inaccuracy. We have revised this point to emphasize the link between recurrent and severe malaria infection and severe malarial anemia in young children. The relevant part of the discussion has been revised as follows.

This choice reflects the importance of *P. falciparum* malaria – the dominant species prevalent in sub-Saharan Africa – as a major etiology of anemia in sub-Saharan Africa. Anemia due to *P. falciparum* infection is likely to be the result of either recurrent infection or the progression of untreated infection to severe malaria. Severe malaria has been shown to present as severe malarial anemia in young children in areas of high malaria incidence [36, 37].

- Line 246-248: I do not follow how non-linearity with covariates indicates that the effect on anaemia is dependant on those covariates, surely dependence is not subject to the shape of the association and the the comparison of the model performance with and without the covariates not a better indicator of this? You say non-linear associations improved the model, did you fit it enforcing linear relationships or is this an assumption?

Response Thanks for raising this clarification question. Penalized splines are a regularized approach, which means that any arbitrary functional form – e.g., linear, nonlinear – can be approximated in a data-driven manner. This ensures that the relationship between any given covariate and the variable of interest is correctly specified while providing a high degree of flexibility. For some covariates (e.g., the age of the children) we found a nonlinear association with anemia. Imposing a linear relationship – in the presence of a non-linear association – would result in a misspecified model. This is illustrated in Figure 3, where on the left-hand side the modeled relationship between the child’s age and the Hb level is shown using a flexible approach based on penalized splines. While the right-hand side shows the result enforcing a linear relationship, which means that for any given age the effect on the Hb level remains constant. Note that on the left-hand side the slope becomes flatter with increasing age, emphasizing that the effect on anemia depends on the level of the covariate (e.g., the effect is different for a 20-month-old child compared to a 45-month-old child). The better fit of the more flexible model is also indicated by the lower deviance information criterion (DIC).

Fig. 3: Estimated effect on the mean Hb level assuming a non-linear association with the age of the child (left) and, effect on the mean Hb level assuming a linear association with the age of the child to illustrate the impact of model misspecification. In addition to the mean effects, both figures include 95% credible intervals. Both models use the training data for the regional sub sample of Madagascar.

- Line 258-260: how do your results compare to Stevens et al. [13] (Ref 34 in this manuscript) who produce national estimates of childhood anaemia. Please clarify that these are the first subnational estimates of anaemia for these regions, not the first.

Response The revised manuscript now includes in the Supplementary Materials the model-based estimates aggregated to the country level. In addition, these tables (*Table S 6.2 to S 6.5*) also include the WHO estimates [13, 14] allowing for an easy comparison. In addition, to be able to get an idea of the burden, these tables also include the UN World Population Prospects [29], allowing a quick calculation of the projected number of affected children.

- Line 264-5: How would these estimates contribute to an early warning system of food insecurity? Whilst I agree this is a good tool for identifying areas requiring support, it is more a long term tool and useful for predictions in areas where measurements have not been done/are not possible. I don't think it works as an early warning system as (a) the food insecurity will be coming before the anaemia, and (b) the input data of anaemia measurement are time consuming and expensive to undertake therefore not regularly updated, and (c) for an early warning system you would be looking for sudden changes in your estimates, if you don't have the input data regularly updated than this would be reliant on your covariates which would need to accurately reflect shocks; the included covariates appear annual at best.

Response The reviewer is correct. We wanted to emphasize that these estimates can help to improve the monitoring of the Sustainable Development Goals and to better target interventions and allocate scarce resources more effectively. We have reworded this as follows:

These insights can help to identify areas where interventions and resources are most urgently needed and can support the improvement of monitoring systems for the SDGs, particularly SDG 2.2 – *Malnutrition: End all forms of malnutrition.*

We sincerely hope that we have addressed all of your remaining concerns, and we apologize again for any oversights on our part. We greatly appreciate your insightful feedback and are confident that the manuscript has improved significantly as a result. Thank you for your valuable input.

References

- [1] Umlauf, N. & Kneib, T. A primer on Bayesian distributional regression. *Statistical Modelling* **18**, 219–247 (2018). doi: [10.1177/1471082X18759140](https://doi.org/10.1177/1471082X18759140).
- [2] Köhler, M., Umlauf, N. & Greven, S. Nonlinear association structures in flexible Bayesian additive joint models. *Statistics in Medicine* **37**, 4771–4788 (2018). doi: [10.1002/sim.7967](https://doi.org/10.1002/sim.7967).
- [3] Lang, S., Umlauf, N., Wechselberger, P., Harttgen, K. & Kneib, T. Multilevel structured additive regression. *Statistics and Computing* **24**, 223–238 (2014). doi: [10.1007/s11222-012-9366-0](https://doi.org/10.1007/s11222-012-9366-0).
- [4] Klein, N., Kneib, T., Klasen, S. & Lang, S. Bayesian structured additive distributional regression for multivariate responses. *Journal of the Royal Statistical Society: Series C (Applied Statistics)* **64**, 569–591 (2015). doi: [10.1111/rssc.12090](https://doi.org/10.1111/rssc.12090).
- [5] Umlauf, N., Klein, N., Simon, T. & Zeileis, A. **bamlss**: A lego toolbox for flexible Bayesian regression (and beyond). *Journal of Statistical Software* **100**, 1–53 (2021). doi: [10.18637/jss.v100.i04](https://doi.org/10.18637/jss.v100.i04).
- [6] Umlauf, N., Klein, N. & Zeileis, A. BAMLSS: Bayesian additive models for location, scale, and shape (and beyond). *Journal of Computational and Graphical Statistics* **27**, 612–627 (2018). doi: [10.1080/10618600.2017.1407325](https://doi.org/10.1080/10618600.2017.1407325).
- [7] Roberts, D. R. *et al.* Cross-validation strategies for data with temporal, spatial, hierarchical, or phylogenetic structure. *Ecography* **40**, 913–929 (2017). doi: [10.1111/ecog.02881](https://doi.org/10.1111/ecog.02881).
- [8] Gething, P. W. *et al.* A new world malaria map: *Plasmodium falciparum* endemicity in 2010. *Malaria Journal* **10**, 378 (2011). doi: [10.1186/1475-2875-10-378](https://doi.org/10.1186/1475-2875-10-378).
- [9] Umlauf, N. *et al.* Scalable estimation for structured additive distributional regression. *Journal of Computational and Graphical Statistics*, accepted on July 25, 2024 (2024). Preprint available under doi: [10.48550/ARXIV.2301.05593](https://doi.org/10.48550/ARXIV.2301.05593) [Computation (stat.CO); Machine Learning (stat.ML)].
- [10] Hastie, T. J. & Tibshirani, R. J. Varying-coefficient models. *Journal of the Royal Statistical Society. Series B (Methodological)* **55**, 757–796 (1993). <http://www.jstor.org/stable/2345993>.
- [11] R Core Team. R: A language and environment for statistical computing (2023). <https://www.R-project.org/>.
- [12] Kaombe, T. M. & Hamuza, G. A. Impact of ignoring sampling design in the prediction of binary health outcomes through logistic regression: evidence from Malawi demographic and health survey under-five mortality data; 2000-2016. *BMC Public Health* **23**, 1674 (2023). doi: [10.1186/s12889-023-16544-4](https://doi.org/10.1186/s12889-023-16544-4).
- [13] Stevens, G. A. *et al.* National, regional, and global estimates of anaemia by severity in women and children for 2000–19: a pooled analysis of population-representative data. *Lancet Global Health* **10**, e627–e639 (2022). doi: [10.1016/S2214-109X\(22\)00084-5](https://doi.org/10.1016/S2214-109X(22)00084-5).
- [14] WHO Global Health Observatory. Prevalence of anaemia in children aged 6–59 months (%) [Dataset] (2022). Retrieved May 13, 2024, from [https://www.who.int/data/gho/data/indicators/indicator-details/GHO/prevalence-of-anaemia-in-children-under-5-years-\(-\)](https://www.who.int/data/gho/data/indicators/indicator-details/GHO/prevalence-of-anaemia-in-children-under-5-years-(-)).
- [15] Murphy, A. H. Skill scores based on the mean square error and their relationships to the correlation coefficient. *Monthly Weather Review* **116**, 2417–2424 (1988). doi: [10.1175/1520-0493\(1988\)116;2417:SSBOTM;2.0.CO;2](https://doi.org/10.1175/1520-0493(1988)116;2417:SSBOTM;2.0.CO;2).
- [16] Gneiting, T. & Raftery, A. E. Strictly proper scoring rules, prediction, and estimation. *Journal of the American Statistical Association* **102**, 359–378 (2007). doi: [10.1198/016214506000001437](https://doi.org/10.1198/016214506000001437).

- [17] Balarajan, Y., Ramakrishnan, U., Özaltın, E., Shankar, A. H. & Subramanian, S. V. Anaemia in low-income and middle-income countries. *Lancet* **378**, 2123–2135 (2011). doi: [10.1016/S0140-6736\(10\)62304-5](https://doi.org/10.1016/S0140-6736(10)62304-5).
- [18] Soares Magalhães, R. J. & Clements, A. C. A. Spatial heterogeneity of haemoglobin concentration in preschool-age children in sub-Saharan Africa. *Bulletin of the World Health Organization* **89**, 459–468 (2011). doi: [10.2471/BLT.10.083568](https://doi.org/10.2471/BLT.10.083568).
- [19] Kassebaum, N. J. *et al.* A systematic analysis of global anemia burden from 1990 to 2010. *Blood* **123**, 615–624 (2014). doi: [10.1182/blood-2013-06-508325](https://doi.org/10.1182/blood-2013-06-508325).
- [20] Brooker, S. *et al.* The co-distribution of *Plasmodium falciparum* and hookworm among African schoolchildren. *Malaria Journal* **5**, 99 (2006). doi: [10.1186/1475-2875-5-99](https://doi.org/10.1186/1475-2875-5-99).
- [21] Karagiannis-Voules, D.-A. *et al.* Spatial and temporal distribution of soil-transmitted helminth infection in sub-Saharan Africa: a systematic review and geostatistical meta-analysis. *Lancet Infectious Diseases* **15**, 74–84 (2015). doi: [10.1016/S1473-3099\(14\)71004-7](https://doi.org/10.1016/S1473-3099(14)71004-7).
- [22] Weaver, H. J., Hawdon, J. M. & Hoberg, E. P. Soil-transmitted helminthiases: implications of climate change and human behavior. *Trends in Parasitology* **26**, 574–581 (2010). doi: [10.1016/j.pt.2010.06.009](https://doi.org/10.1016/j.pt.2010.06.009).
- [23] Kokaliaris, C. *et al.* Effect of preventive chemotherapy with praziquantel on schistosomiasis among school-aged children in sub-Saharan Africa: a spatiotemporal modelling study. *Lancet Infectious Diseases* **22**, 136–149 (2022). doi: [10.1016/S1473-3099\(21\)00090-6](https://doi.org/10.1016/S1473-3099(21)00090-6).
- [24] Lai, Y.-S. *et al.* Spatial distribution of schistosomiasis and treatment needs in sub-Saharan Africa: a systematic review and geostatistical analysis. *Lancet Infectious Diseases* **15**, 927–940 (2015). doi: [10.1016/S1473-3099\(15\)00066-3](https://doi.org/10.1016/S1473-3099(15)00066-3).
- [25] Gayawan, E., Arogundade, E. D. & Adebayo, S. B. Possible determinants and spatial patterns of anaemia among young children in Nigeria: a Bayesian semi-parametric modelling. *International Health* **6**, 35–45 (2014). doi: [10.1093/inthealth/ih034](https://doi.org/10.1093/inthealth/ih034).
- [26] Ngwira, A. & Kazembe, L. N. Analysis of severity of childhood anemia in Malawi: a Bayesian ordered categories model. *Open Access Medical Statistics* **6**, 9–20 (2016). doi: [10.2147/OAMS.S95159](https://doi.org/10.2147/OAMS.S95159).
- [27] Pasricha, S.-R., Colman, K., Centeno-Tablante, E., Garcia-Casal, M.-N. & Peña-Rosas, J.-P. Revisiting WHO haemoglobin thresholds to define anaemia in clinical medicine and public health. *Lancet Haematology* **5**, e60–e62 (2018). doi: [10.1016/S2352-3026\(18\)30004-8](https://doi.org/10.1016/S2352-3026(18)30004-8).
- [28] Croft, T. N., Marshall, A. M. J., Allen, C. K., Assaf, S. & Balian, S. Guide to DHS Statistics (2018). Retrieved April 16, 2018, from https://www.dhsprogram.com/pubs/pdf/DHSG1/Guide_to_DHS_Statistics_DHS-7_v2.pdf.
- [29] United Nations Population Division. World population prospects 2022 [Dataset] (2022). Retrieved May 13, 2024, from <https://population.un.org/wpp/Download/Standard/Population/>.
- [30] Mohammed, S. H., Habtewold, T. D. & Esmailzadeh, A. Household, maternal, and child related determinants of hemoglobin levels of Ethiopian children: hierarchical regression analysis. *BMC Pediatrics* **19** (2019). doi: [10.1186/s12887-019-1476-9](https://doi.org/10.1186/s12887-019-1476-9).
- [31] Ngwira, A. & Kazembe, L. N. Bayesian random effects modelling with application to childhood anaemia in Malawi. *BMC Public Health* **15**, 161 (2015). doi: [10.1186/s12889-015-1494-y](https://doi.org/10.1186/s12889-015-1494-y).
- [32] Roberts, D. J., Matthews, G., Snow, R. W., Zewotir, T. & Sartorius, B. Investigating the spatial variation and risk factors of childhood anaemia in four sub-Saharan African countries. *BMC Public Health* **20**, 126 (2020). doi: [10.1186/s12889-020-8189-8](https://doi.org/10.1186/s12889-020-8189-8).

- [33] Soares Magalhães, R. J. & Clements, A. C. A. Mapping the risk of anaemia in preschool-age children: the contribution of malnutrition, malaria, and helminth infections in west Africa. *PLOS Medicine* **8**, e1000438 (2011). doi: [10.1371/journal.pmed.1000438](https://doi.org/10.1371/journal.pmed.1000438).
- [34] Moschovis, P. P. *et al.* Individual, maternal and household risk factors for anaemia among young children in sub-Saharan Africa: a cross-sectional study. *BMJ Open* **8**, e019654 (2018). doi: [10.1136/bmjopen-2017-019654](https://doi.org/10.1136/bmjopen-2017-019654).
- [35] Kinyoki, D. *et al.* Anemia prevalence in women of reproductive age in low- and middle-income countries between 2000 and 2018. *Nature Medicine* **27**, 1761–1782 (2021). doi: [10.1038/s41591-021-01498-0](https://doi.org/10.1038/s41591-021-01498-0).
- [36] White, N. J. Anaemia and malaria. *Malaria Journal* **17**, 371 (2018). doi: [10.1186/s12936-018-2509-9](https://doi.org/10.1186/s12936-018-2509-9).
- [37] Cohee, L. M. *et al.* Preventive malaria treatment among school-aged children in sub-Saharan Africa: a systematic review and meta-analyses. *Lancet Global Health* (2020). doi: [10.1016/S2214-109X\(20\)30325-9](https://doi.org/10.1016/S2214-109X(20)30325-9).

Rebuttal Letter to *Comments Raised by the Reviewers* for the Revised Article
*High-resolution spatial prediction of anemia risk among children between 6 and
59 months in low- and middle-income countries (COMMSMED-24-0035A)*

submitted to

Communications Medicine

November 15, 2024

Comments Reviewer #1

1 Introduction & Summary of the Content

This is a peer-review of the revised "High-resolution spatial prediction of anemia risk among children between 6 and 59 months in low- and middle-income countries" by Seiler, Wetscher, Harttgen, Utzinger, and Umlauf, submitted Aug 2024.

2 Summary of the Content

The authors note a need for high resolution (spatial-temporal) estimates of anemia in children to help monitor progress in anemia reduction and target precision interventions. Further, they note that existing approaches of this specific outcome are limited to individual or small selections of countries. There is good reason to want to model larger swathes of countries concurrently: namely this allows the modeler to leverage the entire, larger dataset to model associations between the covariates and anemia that are shared across all countries.

To do this, they employ "full probabilistic Bayesian distributional regression models" to estimate the distribution of hemoglobin (Hb) in children of different ages and sexes across a number of low and middle-income countries and for every year from 2005 to 2021. They use flexible distributional models to account for important characteristics (skew) of the Hb distribution that cannot be well-modeled by more often used Normal distributions, and they perform model comparisons to validate the predictive performance of their selected models. They conclude with a brief discussion of some of the results.

3 Overall Revision Feedback

I thank the authors for working so diligently to address my comments and those from the other reviewers. It looks as if it took substantial effort, but I agree with their assessment that the paper is now much improved. Congratulations.

I would like to reiterate two small (but I believe important) requests from my initial set of feedback which were not completely addressed in the revision.

Response We thank Reviewer #1 very much indeed for the critical but positive overall assessment of our revised manuscript and the correct summary of our manuscript entitled *High-resolution spatial prediction of anemia risk among children between 6 and 59 months in low- and middle-income countries (COMMSMED-24-0035A)*. Please find below our point-by-point response to the issues and questions you and two other anonymous reviewers raised.

4 Major Remarks

My line numbers reference the combined marked up pdf document.

1. I appreciate the extra modeling clarity that the authors have included in the revision. I still have some questions about the priors used in the modeling framework and I think it is important that the authors completely specify their model choice, including specifying the details of their priors.

Response Thank you for your thorough review. We have now provided more details on the modeling approach, and in particular on the specification of priors in the supplementary materials. Below is our point-by-point response to your specific comments.

- (a) Priors are now mentioned in the supplement and their first mention appears at line 1065 under the subsection "Re-estimate the sparse model." Is a full Bayesian model not used in the initial covariate selection model fits? If not, how do those models differ from the final model and what are the implications of using different models during variable selection vs the final model?

Response We thank Reviewer #1 for pointing out this inaccuracy. To make our chosen approach transparent, we would like to briefly review the five steps of the modeling strategy that we used:

1. Creating the training and test data sets
2. Reducing the number of suitable candidate distributions
3. Defining candidate predictors η_k for each distributional parameter k
4. Re-estimating the final models omitting uninformative covariates
5. Model evaluation and calibration checks

Accordingly, Reviewer #1 is correct in noting that only the final model is estimated using Bayesian simulation techniques (i.e., Markov chain Monte Carlo (MCMC)). The selection model is estimated using the novel batchwise backfitting algorithm developed by members of our research group and recently published in the *Journal of Computational and Graphical Statistics* [1], including the covariates/model terms specified in Equation 3 of the manuscript. This allows us to identify only those covariates that are considered informative by the algorithm. The final models are then estimated omitting the covariates identified by the algorithm as uninformative. Thus, the model of the selection step and the final model differ only in the way that uninformative covariates have been omitted and the final model is estimated using MCMC sampling. The specified distribution, parameters, and the individual arguments of each included model term were not changed between the estimation steps.

Since the term *Re-estimate the sparse model* can be misinterpreted, we have renamed the fourth step to *Re-estimating the final models omitting uninformative covariates*. In addition, we will provide all R-scripts of the individual modeling steps in the supplementary materials. Note also that the R-script labeled **example.R** included in the supplementary materials illustrates the five modeling steps based on simulated data.

- (b) Line 1065 says "a common choice of priors for linear effects are uninformative normal priors a common choice of priors for linear effects are uninformative normal priors for γ , i.e., $p_{jk}(\gamma_{lk}|y) \propto \text{const.}$ " I have a few concerns with this.

Response We thank Reviewer #1 for raising this point. You are correct that the original statement could be misleading without additional clarification. We have now provided more detailed information on the choice of priors for linear effects and the conditions under which an uninformative normal prior can be considered "flat" or "constant."

- i. First, I would not call a flat/constant prior normal.

Response We appreciate Reviewer #1 for pointing out this ambiguity, and we apologize for any confusion caused. The wording has been revised for clarity. We have elaborated on the prior choice for linear effects, and the revised text now better reflects when an uninformative normal prior can be considered a "flat" prior, including the subtleties of the distinction. The revised section reads as follows:

In this context, a common choice of priors for linear effects are uninformative normal priors for γ , i.e., $p_{jk}(\gamma_{jk}|y) \sim \mathcal{N}(\mathbf{m}, \mathbf{M})$, where \mathbf{m} is the prior mean and \mathbf{M} is the prior covariance matrix. To ensure the prior is uninformative, a typical approach is to increase the variances in \mathbf{M} , or equivalently, to allow the precision matrix \mathbf{M}^{-1} to approach zero. As the variance components in \mathbf{M} become very large (or equivalently, as $\mathbf{M}^{-1} \rightarrow 0$), the prior becomes increasingly flat and resembles a uniform distribution in the limit. However, it's important to note that a normal prior with very large variance does not strictly become a uniform distribution, but rather a Gaussian distribution with a very broad spread. A key consideration is whether the uninformative prior is parameterized by the standard deviation σ or the variance σ^2 . The choice of variance σ^2 , as used here, ensures that the prior is weakly informative and very flat. This results in a proper, but uninformative, prior for the linear effects, which does not overly constrain the model. For further discussion of the choice of prior for linear effects, see, e.g., [2, 3].

- ii. Second, "flat" priors are not synonymous with uninformative priors (nor are they invariant to parameter transformations). The common example is that a flat prior on a standard deviation from a Gaussian distribution, σ , puts a lot of prior weight of σ being far from zero, ie it is actually a strong prior on having a large value of σ .

Response We appreciate this insightful comment from Reviewer #1, and we apologize for the potential confusion in our initial explanation. We have revised this paragraph (see response above) to make it clearer when an uninformative normal prior can be considered a flat prior, and we've addressed the difference between "flat" and "uninformative" priors in more detail. Thank you for helping us clarify this distinction.

- iii. Third, constant priors are not proper distributions and there is no guarantee that the posteriors will be proper. This seems at odds with the authors' claims (in both the rebuttal letter and on line 1069) that "the primary reason we use a Bayesian model is not to incorporate prior knowledge to improve the model but to achieve valid inference." I understand the sentiment of their comment (in fact, I often use Bayesian models for the same reason), but if valid inference is the primary reason for employing Bayesian methods, then why not use proper Bayesian priors?

Response We thank Reviewer #1 for raising this important point and for helping us further clarify our approach. To address the concern: while constant priors are indeed improper, we would like to clarify that the priors we have used are proper priors. While they are weakly informative and may appear "flat" due to large variances, they still satisfy the conditions for a proper prior distribution, which ensures that the resulting posterior distributions are valid and proper. We hope this distinction helps clarify our approach.

There are some real reasons to be hesitant about using improper priors (eg see Marginalization Paradoxes in Bayesian and Structural Inference by Dawid, Stone, and Zidek [4].)

Response We thank Reviewer #1 for mentioning the potential issues with improper priors, as discussed in [4]. However, we would like to clarify that since the priors we have used are proper priors, this concern does not apply in our case. As such, we believe that the use of proper priors in our model ensures that the results are valid and the inference is sound.

- (c) Line 1067 says that, "For non-linear effects, which are typically based on a basis function approach, a multivariate normal prior for β_{lk} , with a model term-specific precision matrix is used." The paragraph concludes with "generic multivariate normal priors for smooth functions have demonstrated 1072 high efficiency and robustness in numerous applications". Do the priors used on the non-linear effects in this paper have "model term-specific precision matrices" or are "generic multivariate normal priors"?

used? Either way, please write down the specific choice of priors used. As my previous comments about the flat prior attempt to convey, the details of the priors used do matter. This is even more important since the complete code for the full model (understandably) is not available to reproduce the results and we have no other way of knowing how the model was completely specified.

Response Thank you for pointing this out. You are absolutely correct that the precision matrices are specific to each model term. We apologize for the earlier ambiguity and appreciate your careful attention to this detail. We have revised the paragraph on the prior selection for nonlinear effects to provide more clarity. The updated version reads as follows:

For non-linear effects, which are typically based on a basis function approach, a multivariate normal prior for the basis function β_{jk} , with a model term-specific precision matrix is used (see, e.g., [2]). Thus the prior $p_{jk}(\cdot)$ of β_{jk} assumes a multivariate normal kernel of the following form: $d_{\beta_{jk}}(\beta_{jk} | \tau_{jk}, \alpha_{\beta_{jk}}) \propto |\mathbf{P}_{jk}(\tau_{jk})|^{-\frac{1}{2}} \exp\left(-\frac{1}{2} \beta_{jk}^\top \mathbf{P}_{jk}(\tau_{jk}) \beta_{jk}\right)$. By specifying the penalty matrices $\alpha_{\beta_{jk}} = \{\mathbf{K}_{1jk}, \dots, \mathbf{K}_{Ljk}\}$ for the basis function coefficients β_{jk} the precision matrix, e.g., for univariate splines $\mathbf{P}_{jk}(\tau_{jk}) = \tau_{1jk}^{-2} \mathbf{K}_{1jk}$, can be obtained, where the penalty matrices are specific to the chosen smooth term specification. For the smoothing variances τ_{jk} a inverse gamma distribution is used for each $\tau_{jk} = (\tau_{1jk}, \dots, \tau_{Ljk})^\top$. In the latter case, smoothing variances, also sampled from the posterior distribution, control the amount of smoothness/overfitting, similar to the frequentist’s smoothing parameter. The primary reason we use a Bayesian model is not to incorporate prior knowledge to improve the model but to achieve valid inference, which is almost impossible to obtain for distributional regression models using the frequentist approach. For details see e.g., [2] for a more detailed description of the prior choice for non-linear effects. Generic multivariate normal priors for smooth functions with a precision matrix \mathbf{P}_{jk} specific to the basis function β_{jk} have demonstrated high efficiency and robustness in numerous applications, (see, e.g., [5–9]).

We have also included the complete R code used to pre-process the data, estimate the models, and predict anemia prevalence. We will continue to ensure that we help interested researchers reproduce our results – or apply similar models to other applications. Please note that we were unable to upload the R scripts directly to the *Communications Medicine* submission platform. Therefore, we have converted the R scripts to PDFs. However, we are happy to provide the R scripts in the supplementary materials to bona fide interested researchers. Note also that all the software used is open source.

2. My other comment still pertains to validation and model presentation. The new clarifications around the validation method and the admin2 level validation results go quite a long way towards assuaging my concerns. Still, I feel there is mismatch between the model resolution and presentation (continuous/pixel) and the validation (discrete admin units). You have a number of good options to resolve this completely, and all pertain to aligning these two concepts or taking a few sentences to explain why they are different.

Response Thank you for bringing this issue to our attention. We agree with the reviewer, that model validation should be performed at the primary level of interest. Therefore, we provide the validation of the continuous hemoglobin (Hb) level at the pixel level using the out-of-sample test data (see also Figure S 15 of the manuscript). Overall, we find the overall fit of the models to be impressive and competitive given the high degree of variability associated with the data. Note also that all diagnostic and calibration checks (see Figures S 5 to S 7 of the manuscript) and the calculation of the corresponding metrics were performed on the out-of-sample data without any data aggregation. In addition, we would like to mention that in the validation process, we also considered approaches commonly used in Global Burden of Disease (GBD) studies (see e.g., [10]). To our surprise, the mentioned study only provides in-sample validation at the admin-0, admin-1, and admin-2 levels in the Supplement, although this is referred to as out-of-sample validation in the main manuscript.

- You could add validation results to the supplement which show how well the model is performing at the continuous/data observation level.

Fig. 1: Scatter plot of the survey-based Hb concentrations reported by DHS and the corresponding model-based estimates aggregated to the pixel level (left) and scatter plot of the survey-based Hb concentrations reported by DHS (calculated using only the out-of-sample data) and the corresponding model-based estimates aggregated to the pixel level (right). In addition, the panel shows 95% credible intervals for each location and the correlation coefficient ρ for each geographic region.

Response The revised version of the manuscript now includes pixel-level validation of the continuous outcome (i.e., the Hb level). See Figure 1 (see also Figure S 15 of the manuscript) for validation of the Hb level at the pixel level using the in-sample data, as well as the out-of-sample data. These figures are also been included in the manuscript. Note also that all diagnostic and calibration checks (see Figures S 5 to S 7 of the manuscript) and the calculation of the corresponding metrics were performed on the out-of-sample data without any data aggregation. Given the high resolution and high variability of the data, we find the overall fit of the models to be impressive and comparable to other studies with similar content.

- If the primary target of inference is admin2 units, then that should be clarified and in this case I think it is also worth discussing why you are using continuous spatial models for discrete spatial outcomes.

Response We now also provide the validation process at the continuous/pixel level. Accordingly, this and the following points should be considered irrelevant.

- You could show your main continental results (Figure 4) at the admin 2 level and not at the pixel level.

Response See previous comment, we now also provide the validation process at the continuous/pixel level. Accordingly, this and the following points should be considered irrelevant.

- etc

Response See previous comment, we now also provide the validation process at the continuous/pixel level. Accordingly, this and the following points should be considered irrelevant.

We would like to thank Reviewer #1 for the overall positive review. Especially the detailed, constructive and yet challenging feedback is very much appreciated. It helped us a lot to improve the current quality of the article. We hope that we have been able to address all of Reviewer #1's remaining points adequately.

Comments Reviewer #2

I went through the Authors' responses on the comments I provided and those raised by other reviewers, reviewed the revised manuscript and all the additions done in the main document and the Supp File. Extensive revision and improvement have been done into the paper.

Specifically:

- The additions in the Abstract, Methods [modelling and calibration] and Results (both in the main manuscript and Supp file) have created clarity in the areas that were ambiguous or hard to follow. I appreciate additions of Country-specific and subnational results in the paper.
- Revisions done to the Figures and Plots is appreciated.
- Revisions in the interpretation of results is sufficiently done.
- If Authors wish – In the Discussion – There was no need to start by mentioning the Limitations of this study, that is a great addition but still can go at the end of the section – before Conclusion.

No further comments for this work.

We thank Reviewer #2 for the positive review and appreciate the detailed, constructive, and sometimes challenging feedback in the first round of reviews, which helped us greatly to improve the current quality of the article. We are pleased that we were able to address all of the points raised in a satisfactory manner.

Comments Reviewer #3

Summary

I thank the authors for the revisions to the manuscript "High-resolution spatial prediction of anaemia risk among children between 6 and 59 months in low-and middle-income countries". The revision have addressed my initial comments well and greatly improved the manuscript, both in terms of readability, strength of the results and utility/interpretation of the results. The addition of Figure 3, extension of the discussion of the results and model validation are particularly beneficial.

Response We thank Reviewer #3 for his/her overall positive and constructive feedback. Below is our point-by-point response to the open questions, comments, and ambiguities raised during the review.

Specific comments

I just have a couple of remaining points/queries that I think would be beneficial to address. The modelled estimates in some countries appear to have changed quite a lot from the previous submission. The countries I have concerns about have very different estimates than in WHO and appear are a bit inconsistent with my preconceptions of general health levels in these countries. These countries have no data from DHS so I would like to understand what is driving these estimates:

Response Thank you for pointing us to this important aspect, however, please note that we have been asked in the first round of review to alter our used color palette. Thus from our perspective, a direct quantitative comparison between the estimates of the initial submission and the estimates of the revised version is not meaningful due to the differing color palette. Accordingly, the estimates can only be assessed qualitatively, if at all. However, we completely agree with Reviewer #3 that the differences in the magnitude between the WHO estimates and our estimates need further attention. For this, we conducted an extensive literature review searching for anemia prevalence estimates for those specific countries. The results of this extensive literature review are summarized in Table 1 (see also Table S 8 of the manuscript). We briefly want to highlight important aspects: First, in any case the anemia prevalence is estimated to be well above 40% (the WHO cut-off to classify a serious public health problem). Second, these discrepancies highlight from our perspective the need for a more accurate surveillance that includes more frequent georeferenced surveys. Without this data the estimates rely solely on the information based on the estimated association between the included covariates and the Hb level, and the model has been trained without these specific countries. We also added this to the discussion which now reads as follows:

In contrast, the Stevens [11] and WHO estimates are more conservative for countries such as Chad, the Central African Republic, or South Sudan. These are also countries for which recent DHS data with Hb measurements are not available, and thus our model predicts anemia prevalence based on the included covariates and the estimated spatio-temporal association. This implies that the model-based associations between the included covariates and Hb levels, and the observed spatio-temporal dynamics are similar between included countries and countries without direct ground observations. To investigate whether these estimates are nevertheless plausible, a comprehensive literature review aims to examine them more closely for plausibility. For a comprehensive comparison of the estimates, see Table S 8 of the supplementary materials. Overall, the estimates of anemia prevalence at the country level from this study fall within the uncertainty intervals or overlap substantially in most cases. In some cases, however, there are discrepancies between some country-level studies and surveys, the Stevens [11] and WHO estimates, and the model-based estimates presented in this research article. However, this highlights two important aspects that should be emphasized: First, all of the included case reports at the country level that are based on studies or surveys indicate an anemia prevalence well above 40% – i.e., the WHO cut-off for indicating a serious public health problem in this context. This means that anemia is certainly a serious public health problem in these countries. Second, the discrepancies highlight the importance of the need for better and more accurate surveillance, including more frequent geo-referenced surveys. This would allow a better monitoring, and targeting of the population at risk.

Table 1: Comparison of studies on anemia, estimated anemia prevalence of this study, and estimated anemia prevalence of the WHO [39] among children together with 95% uncertainty intervals (if available) of any anemia (i.e., $P(\text{Hb} < 110\text{g L}^{-1})$) in children aged 6–59 months at the country level, 2009–2022.

Country code	Author	Year	Age (months)	Sample size	Comment	Prevalence of study (%)	Prevalence (%)	Prevalence WHO (%)
Articles								
AFG	Fahim et al. [12]	2013	6–59	905	National representative	44.8 [41.5 - 48]	57.1 [51.8 - 62.4]	43.5 [34.4 - 53.4]
AFG	Stanikzai et al. [13]	2021/2022	6–59	512	Kandahar city	55.6 [51.2 - 60]	57.8 [52.4 - 63.1] in 2020	44.9 [28.1 - 64.4] in 2019
BTN	Campbell et al. [14]	2015	6–59	1083	National Nutrition Survey	42.3 [35.8 - 48.8]	63.4 [60.7 - 66]	45.7 [32.7 - 60.8]
BTN	Chhetri et al. [15]	2016	6–59	353	Hospital-based	58.4	63.3 [60.6 - 65.9]	47.2 [35.8 - 60.4]
BWA	Leepile et al. [16]	2019	6–59	367	Indigenous San People	68	62.8 [59.5 - 66.1]	43.3 [21.7 - 67.2]
CAF	Vonaesch et al. [17]	2017/2018	24–60	409	Bangui	47	57.2 [54.7 - 59.6]	75.2 [63.1 - 83.2]
GNB	Thorne et al. [18]	2012	6–59	440	Bijagós Archipelago	80.2 [76.3 - 83.7]	73.6 [72.2 - 75.1]	72.2 [57.2 - 81.7]
GNQ	Ncogo et al. [19]	2013	2–59	350	Bata district	87.7	74.1 [72.3 - 75.9]	67.5 [58.4 - 74.9]
GNQ	Ncogo et al. [19]	2013	13–59	441	Bata district	85.6	74.1 [72.3 - 75.9]	67.5 [58.4 - 74.9]
PAK	Habib et al. [20]	2011/2012	6–59	7138	National Nutrition Survey	62.3 [58.2 - 65.5]	63.9 [60.9 - 66.8]	58.2 [48.7 - 66.8]
PAK	Habib et al. [21]	2018	6–59	17814	National Nutrition Survey	53.7	60.8 [57.8 - 63.8]	53.8 [40.3 - 67.1]
SOM	Wirth et al. [22]	2019	6–59	1667	Somalia Micronutrient Survey	43.4 [40 - 46.9]	78.2 [75.7 - 80.7]	51.8 [39.7 - 65.8]
TCD	Zavala et al. [23]	2016	6–59	9109	National representative	68.6 [67.7 - 69.6]	58 [55.1 - 60.8]	70.9 [63.7 - 76.7]
TCD	Zavala et al. [23]	2021	6–59	6751	National representative	59.6 [58.5 - 60.8]	54.6 [51.5 - 57.6]	66.3 [59.8 - 72.9] in 2019
Surveys								
BDI	UNHCR [24]	2019	6–59	126	Burundi refugee camps (RC), COD	56.5 [52.3 - 60.6]	55.2 [53.6 - 56.8]	58 [46.3 - 68.9]
BTN	UNHCR [25]	2018	6–59	216	Bhutanese RC, NPL	35.7 [27.4 - 44.7]	63.1 [60.5 - 65.7]	45.2 [30.6 - 61.5]
BWA	UNHCR [26]	2013	6–59	165	Dukwi RC, BWA	50.9 [43 - 58.8]	56.3 [53.3 - 59.4]	39.1 [21.4 - 60.2]
CAF	UNHCR [27]	2019	6–59	249	Makpandu RC, SSD	60.3 [54 - 66.4]	54.2 [51.6 - 56.8]	73.6 [59.2 - 83.4]
CAF	UNHCR [28]	2019	6–59	1820	Central African Republic RC, COD	56.4 [54.2 - 58.8]	54.2 [51.6 - 56.8]	73.6 [59.2 - 83.4]
COD	UNHCR [27]	2019	6–59	249	Makpandu RC, SSD	60.3 [54 - 66.4]	63.5 [61.3 - 65.7]	64.9 [50.3 - 76.6]
DJI	WFP [29]	2011	pre-school	–	Djibouti	65.8	74.8 [72.6 - 77]	52.8 [34.3 - 72.1]
ERI	UNHCR [27]	2019	6–59	249	Makpandu RC, SSD	60.3 [54 - 66.4]	53.2 [49.8 - 56.5]	48.8 [29 - 69.7]
MRT	DHS [30]	2019-2021	6–59	10077	National representative	76.7	74.1 [72.1 - 76.1] in 2020	65.5 [47.5 - 78] in 2019
SOM	FSNAU [31]	2009	6–59	784	FSNAU	59.3 [54.8 - 63.6]	78.2 [75.8 - 80.5]	59.3 [48.2 - 69.3]
SOM	UNHCR [32]	2018	6–59	415	Somali RC Bokolmany, ETH	44.8 [40 - 49.8]	78.8 [76.4 - 81.1]	52.7 [41.7 - 65.6]
SOM	UNHCR [32]	2018	6–59	360	Somali RC Melkadida, ETH	45.3 [40.1 - 50.6]	78.8 [76.4 - 81.1]	52.7 [41.7 - 65.6]
SOM	UNHCR [32]	2018	6–59	307	Somali RC Kobe, ETH	60.3 [54.5 - 65.8]	78.8 [76.4 - 81.1]	52.7 [41.7 - 65.6]
SOM	UNHCR [32]	2018	6–59	405	Somali RC Hilaweyn, ETH	44.7 [39.9 - 49.6]	78.8 [76.4 - 81.1]	52.7 [41.7 - 65.6]
SOM	UNHCR [32]	2018	6–59	383	Somali RC Buramino, ETH	47.5 [42.4 - 52.7]	78.8 [76.4 - 81.1]	52.7 [41.7 - 65.6]
SSD	UNHCR [33]	2019	6–59	420	South Sudan RC, COD	66 [61.3 - 70.3]	45.9 [43.2 - 48.6]	60.5 [39.5 - 77.4]
SSD	UNHCR [34]	2019	6–59	249	Gorom RC, SSD	70.7	45.9 [43.2 - 48.6]	60.5 [39.5 - 77.4]
SSD	UNHCR [35]	2019	6–59	258	Pamir RC, SSD	47.3 [41.1 - 53.6]	45.9 [43.2 - 48.6]	60.5 [39.5 - 77.4]
SSD	UNHCR [36]	2019	6–59	611	Doro RC, SSD	55.8 [51.9 - 59.7]	45.9 [43.2 - 48.6]	60.5 [39.5 - 77.4]
SSD	UNHCR [36]	2019	6–59	546	Yusuf Batil RC, SSD	55.7 [51.5 - 59.8]	45.9 [43.2 - 48.6]	60.5 [39.5 - 77.4]
SSD	UNHCR [36]	2019	6–59	678	Gendrassa RC, SSD	57.5 [53.8 - 61.2]	45.9 [43.2 - 48.6]	60.5 [39.5 - 77.4]
SSD	UNHCR [36]	2019	6–59	551	Kaya RC, SSD	49.9 [45.8 - 54.1]	45.9 [43.2 - 48.6]	60.5 [39.5 - 77.4]
TCD	UNHCR [37]	2016	6–59	343	Doholo RC, TCD	76.4 [71.6 - 80.6]	58 [55.1 - 60.8]	70.9 [63.7 - 76.7]
TCD	UNHCR [37]	2016	6–59	464	Dosseye RC, TCD	51.9 [45.6 - 58.3]	58 [55.1 - 60.8]	70.9 [63.7 - 76.7]
TCD	UNHCR [37]	2016	6–59	344	Gondjé RC, TCD	73.5 [67.5 - 79.6]	58 [55.1 - 60.8]	70.9 [63.7 - 76.7]
TCD	UNHCR [37]	2016	6–59	395	Amboko RC, TCD	66.6 [60.5 - 72.7]	58 [55.1 - 60.8]	70.9 [63.7 - 76.7]
TCD	UNHCR [37]	2016	6–59	410	Belom RC, TCD	59.3 [54.3 - 64.2]	58 [55.1 - 60.8]	70.9 [63.7 - 76.7]
TCD	UNHCR [37]	2016	6–59	549	Moyo RC, TCD	48.6 [42.8 - 54.4]	58 [55.1 - 60.8]	70.9 [63.7 - 76.7]
TCD	UNHCR [37]	2016	6–59	483	Daressalam RC, TCD	54.5 [49.2 - 59.7]	58 [55.1 - 60.8]	70.9 [63.7 - 76.7]
TCD	UNHCR [38]	2021	6–59	10072	Ensemble RC, TCD	48.1 [46.5 - 49.8]	58 [55.1 - 60.8]	70.9 [63.7 - 76.7]

Note that in the Makpandu refugee camp [27] most refugees come from four African countries (i.e., CAF, COD, SDN, and ERI). Country abbreviations correspond to the country-specific ISO3 country codes. Additional sources: [11, 39], and own calculations.

1. Somalia is estimated to have the highest prevalence of anaemia of all countries.

Response Thank you for highlighting this important issue. We have now carefully reviewed the literature for the countries mentioned – indeed, for all countries without direct field data – to see if our findings can be supported by existing country studies. We found the Office of the United Nations High Commissioner for Refugees (UNHCR) Standardised Expanded Nutrition Survey (SENS) data to be one of the most common data sources – although it can only provide a glimpse and not the full picture. The SENS are standardized surveys conducted in refugee settlements. Overall, except for Somalia, these additional numbers show a very close overlap with our aggregate estimates. See Table 1 (Table S 8 in the supplementary material of the article) for a comprehensive summary of our literature review on this topic. In addition, we now discuss these discrepancies in detail in the manuscript and discuss possible reasons for these discrepancies.

2. South Sudan is estimated to have the lowest prevalence of any anaemia. This especially shocked me as you have discussed the importance of household wealth as a covariate in the model.

Response Thank you for bringing these discrepancies to our attention. As mentioned above, we have extended our discussion of these discrepancies in more detail and conducted an extensive literature review to provide more evidence that the observed patterns can be considered plausible. A comprehensive summary of this literature review is provided in Table 1 (Table S 8 in the supplementary material of the article). For South Sudan, as well as the Central African Republic, we find that one of the best data sources is the UNHCR SENS conducted in refugee settlements. Overall, we would argue that there appears to be considerable overlap, but we are aware that the SENS does not provide representative data at the country level, so these surveys can only provide a snapshot.

3. Finally CAR is also a considerably lower anaemia prevalence than I would expect (again, based on HH wealth being an important factor).

Response Thank you for pointing this out. Please see our response to your previous point, which should hopefully address this point as well.

Minor comments:

Introduction

- Line 174: Tables S4-7 are the nationally aggregated estimated of anaemia, not the pixel level estimates as stated.

Response We would like to thank Reviewer #3 for pointing out this ambiguity. We have corrected this error in the revised version.

- Line 203: States tables S4-7 are for years 2020, 2019 and 2020 (typo for first 2020 = 2010).

Response Thanks for spotting this typo, of course we meant 2010 and not 2020. This has been corrected.

- Line 217: "194 million children" – great to have this estimate added, can you add the uncertainty?

Response Thank you for spotting this. We now provide the estimate based on the geo-political region including uncertainty intervals. The sentence has now changed as follows:

The analysis provides evidence of the substantial burden of anemia in children aged 6 and 59 months, which is estimated to affect about 98.7 [94.5-102.8] million children in sub-Saharan Africa and about 95.1 [91.1-99.0] million children in South Asia in 2020.

- Line 295-299: A good start in talking about the differences but as I mentioned, Somalia has no surveys and is coming up with very high levels of anaemia. I am not disputing that this could be true, and I am actually more concerned about the low prevalence of anaemia your model estimates in South Sudan and CAR. Are there any sources which can confirm these patterns? I think this manuscript could benefit from a bit more of a discussion about the disparities between yours and WHO estimates and what is driving them.

Response Thank you for bringing this to our attention. Please see our response to your previous points on this issue. We now provide a more structured discussion of these points in the manuscript and address this issue with an extensive literature review (see Table 1 for a comprehensive summary). In particular, two points should be added that we now also include in the discussion as follows:

First, all of the included case reports at the country level that are based on studies or surveys indicate an anemia prevalence well above 40% – i.e., the WHO cut-off for indicating a serious public health problem in this context. This means that anemia is certainly a serious public health problem in these countries. Second, the discrepancies highlight the importance of the need for better and more accurate surveillance, including more frequent geo-referenced surveys. This would allow a better monitoring, and targeting of the population at risk.

We sincerely hope that we have addressed all of your remaining concerns, and we apologize again for any oversights on our part. We greatly appreciate your insightful feedback and are confident that the manuscript has improved significantly as a result. Thank you for your valuable input.

References

- [1] Umlauf, N. *et al.* Scalable estimation for structured additive distributional regression. *Journal of Computational and Graphical Statistics* **0**, 1–23 (2024). doi: [10.1080/10618600.2024.2388604](https://doi.org/10.1080/10618600.2024.2388604).
- [2] Umlauf, N., Klein, N. & Zeileis, A. BAMLSS: Bayesian additive models for location, scale, and shape (and beyond). *Journal of Computational and Graphical Statistics* **27**, 612–627 (2018). doi: [10.1080/10618600.2017.1407325](https://doi.org/10.1080/10618600.2017.1407325).
- [3] Fahrmeir, L., Kneib, T., Lang, S. & Brian, M. *Regression: Models, Methods and Applications* (Springer Berlin, Heidelberg, 2021), 2 edn. doi: [10.1007/978-3-662-63882-8](https://doi.org/10.1007/978-3-662-63882-8).
- [4] Dawid, A. P., Stone, M. & Zidek, J. V. Marginalization Paradoxes in Bayesian and Structural Inference. *Journal of the Royal Statistical Society: Series B (Methodological)* **35**, 189–213 (1973). doi: [10.1111/j.2517-6161.1973.tb00952.x](https://doi.org/10.1111/j.2517-6161.1973.tb00952.x).
- [5] Lang, S., Umlauf, N., Wechselberger, P., Harttgen, K. & Kneib, T. Multilevel structured additive regression. *Statistics and Computing* **24**, 223–238 (2014). doi: [10.1007/s11222-012-9366-0](https://doi.org/10.1007/s11222-012-9366-0).
- [6] Klein, N., Kneib, T., Klasen, S. & Lang, S. Bayesian structured additive distributional regression for multivariate responses. *Journal of the Royal Statistical Society: Series C (Applied Statistics)* **64**, 569–591 (2015). doi: [10.1111/rssc.12090](https://doi.org/10.1111/rssc.12090).
- [7] Köhler, M., Umlauf, N. & Greven, S. Nonlinear association structures in flexible Bayesian additive joint models. *Statistics in Medicine* **37**, 4771–4788 (2018). doi: [10.1002/sim.7967](https://doi.org/10.1002/sim.7967).
- [8] Umlauf, N. & Kneib, T. A primer on Bayesian distributional regression. *Statistical Modelling* **18**, 219–247 (2018). doi: [10.1177/1471082X18759140](https://doi.org/10.1177/1471082X18759140).
- [9] Umlauf, N., Klein, N., Simon, T. & Zeileis, A. **bamlss**: A lego toolbox for flexible Bayesian regression (and beyond). *Journal of Statistical Software* **100**, 1–53 (2021). doi: [10.18637/jss.v100.i04](https://doi.org/10.18637/jss.v100.i04).
- [10] Kinyoki, D. *et al.* Anemia prevalence in women of reproductive age in low- and middle-income countries between 2000 and 2018. *Nature Medicine* **27**, 1761–1782 (2021). doi: [10.1038/s41591-021-01498-0](https://doi.org/10.1038/s41591-021-01498-0).
- [11] Stevens, G. A. *et al.* National, regional, and global estimates of anaemia by severity in women and children for 2000–19: a pooled analysis of population-representative data. *Lancet Global Health* **10**, e627–e639 (2022). doi: [10.1016/S2214-109X\(22\)00084-5](https://doi.org/10.1016/S2214-109X(22)00084-5).
- [12] Fahim, O. *et al.* Double burden of malnutrition in Afghanistan: Secondary analysis of a national survey. *PLOS ONE* **18**, 1–19 (2023). doi: [10.1371/journal.pone.0284952](https://doi.org/10.1371/journal.pone.0284952).
- [13] Stanikzai, M. H., Zakir, S., Ishaq, N. & Rahimi, B. A. Prevalence of anemia and its associated factors among children under 5 years of age attending a comprehensive healthcare facility in Kandahar city, Afghanistan. *Indian Journal of Public Health* **66** (2022). doi: [10.4103/ijph.ijph.2202.210](https://doi.org/10.4103/ijph.ijph.2202.210).
- [14] Campbell, R. K. *et al.* Epidemiology of anaemia in children, adolescent girls, and women in Bhutan. *Maternal & Child Nutrition* **14**, e12740 (2018). doi: [10.1111/mcn.12740](https://doi.org/10.1111/mcn.12740).
- [15] Chhetri, K., Mynak, M. L. & Pedon, K. Anemia and risk factors among children 6 months to 59 months old: a hospital-based prospective study. *Bhutan Health Journal* **3**, 1–4 (2017). doi: [10.47811/bhj.45](https://doi.org/10.47811/bhj.45).
- [16] Leepile, T. T. *et al.* Anemia prevalence and anthropometric status of indigenous women and young children in rural Botswana: The San People. *Nutrients* **13** (2021). doi: [10.3390/nu13041105](https://doi.org/10.3390/nu13041105).
- [17] Vonaesch, P. *et al.* Factors associated with stunted growth in children under five years in Antananarivo, Madagascar and Bangui, Central African Republic. *Maternal and Child Health Journal* **25**, 1626–1637 (2021). doi: [10.1007/s10995-021-03201-8](https://doi.org/10.1007/s10995-021-03201-8).

- [18] Thorne, C. J. *et al.* Anaemia and malnutrition in children aged 0–59 months on the Bijagós Archipelago, Guinea-Bissau, West Africa: a cross-sectional, population-based study. *Paediatrics and International Child Health* **33**, 151–160 (2013). doi: [10.1179/2046905513Y.0000000060](https://doi.org/10.1179/2046905513Y.0000000060).
- [19] Ncogo, P. *et al.* Prevalence of anemia and associated factors in children living in urban and rural settings from Bata District, Equatorial Guinea, 2013. *PLOS ONE* **12**, 1–14 (2017). doi: [10.1371/journal.pone.0176613](https://doi.org/10.1371/journal.pone.0176613).
- [20] Habib, M. A. *et al.* Prevalence and predictors of iron deficiency anemia in children under five years of age in Pakistan, a secondary analysis of national nutrition survey data 2011-2012. *PLOS ONE* **11**, 1–13 (2016). doi: [10.1371/journal.pone.0155051](https://doi.org/10.1371/journal.pone.0155051).
- [21] Habib, A. *et al.* Prevalence and risk factors for iron deficiency anemia among children under five and women of reproductive age in Pakistan: Findings from the national nutrition survey 2018. *Nutrients* **15** (2023). doi: [10.3390/nu15153361](https://doi.org/10.3390/nu15153361).
- [22] Wirth, J. P. *et al.* Risk factors of anaemia and iron deficiency in Somali children and women: Findings from the 2019 Somalia micronutrient survey. *Maternal & Child Nutrition* **18**, e13254 (2022). URL <https://onlinelibrary.wiley.com/doi/abs/10.1111/mcn.13254>.
- [23] Zavala, E., Adler, S., Wabyona, E., Ahimbisibwe, M. & Doocy, S. Trends and determinants of anemia in children 6-59 months and women of reproductive age in Chad from 2016 to 2021. *BMC Nutrition* **9**, 117 (2023). doi: [10.1186/s40795-023-00777-y](https://doi.org/10.1186/s40795-023-00777-y).
- [24] UN Refugee Agency. Congo, Dem. Rep.: Standardised Expanded Nutrition Survey (refugees from Burundi, Central African Republic and South Sudan) (2019). Retrieved November 4, 2024, from <https://microdata.unhcr.org/index.php/catalog/782/download/2718>.
- [25] UN Refugee Agency. Nepal: Standardised Expanded Nutrition Survey (Bhutanese Refugee Camps) (2019). Retrieved November 7, 2024, from <https://microdata.unhcr.org/index.php/catalog/521/download/2013>.
- [26] UN Refugee Agency. Botswana: Standardised Expanded Nutrition Survey (Dukwi) (2013). Retrieved November 4, 2024, from <https://microdata.unhcr.org/index.php/catalog/663/download/2429>.
- [27] UN Refugee Agency. South Sudan: Standardised Expanded Nutrition Survey (Makpandu refugee camps) (2019). Retrieved November 4, 2024, from <https://microdata.unhcr.org/index.php/catalog/596/download/2185>.
- [28] UN Refugee Agency. Congo, Dem. Rep.: Standardised Expanded Nutrition Survey (refugees from Burundi, Central African Republic and South Sudan) (2019). Retrieved November 4, 2024, from <https://microdata.unhcr.org/index.php/catalog/782/download/2717>.
- [29] World Food Programme. Urban in-depth EFSA Djibouti (2011). Retrieved October 31, 2024, from <https://citeseerx.ist.psu.edu/document?repid=rep1&type=pdf&doi=8fbffed2efa0a5883a022756f7da1bfd64a49cb1>.
- [30] Office Nationale de la Statistique (ONS), Ministère de la Santé (MS) & ICF. Enquête Démographique et de Santé en Mauritanie 2019-2021: rapport de synthèse (2011). Retrieved October 31, 2024, from <https://dhsprogram.com/pubs/pdf/FR373/FR373.pdf>.
- [31] FSNAU, FAO & UCL. National Micronutrient and Anthropometric Nutrition Survey Somalia (2009). Retrieved November 6, 2024, from <https://fsnau.org/downloads/Somalia-National-Micronutrient-Study.pdf>.
- [32] UN Refugee Agency. Ethiopia: Standardised Expanded Nutrition Survey (Melkadida) (2018). Retrieved November 6, 2024, from <https://microdata.unhcr.org/index.php/catalog/114/download/331>.

- [33] UN Refugee Agency. Congo, Dem. Rep.: Standardised Expanded Nutrition Survey (refugees from Burundi, Central African Republic and South Sudan) (2019). Retrieved November 4, 2024, from <https://microdata.unhcr.org/index.php/catalog/782/download/2719>.
- [34] UN Refugee Agency. South Sudan: Standardised Expanded Nutrition Survey (Gorom refugee camp) (2019). Retrieved November 4, 2024, from <https://microdata.unhcr.org/index.php/catalog/584/download/2194>.
- [35] UN Refugee Agency. South Sudan: Standardised Expanded Nutrition Survey (Jamjang refugee camp) (2019). Retrieved November 4, 2024, from <https://microdata.unhcr.org/index.php/catalog/590/download/2168>.
- [36] UN Refugee Agency. South Sudan: Standardised Expanded Nutrition Survey (Maban refugee camps) (2019). Retrieved November 4, 2024, from <https://microdata.unhcr.org/index.php/catalog/595/download/2179>.
- [37] UN Refugee Agency. Chad: Standardised Expanded Nutrition Survey (South, South-East and West) (2016). Retrieved November 4, 2024, from <https://microdata.unhcr.org/index.php/catalog/665/download/2427>.
- [38] UN Refugee Agency. Chad: Standardised Expanded Nutrition Survey (East, South and Lake Region) (2021). Retrieved November 4, 2024, from <https://microdata.unhcr.org/index.php/catalog/775/download/2697>.
- [39] WHO Global Health Observatory. Prevalence of anaemia in children aged 6–59 months (%) [Dataset] (2022). Retrieved May 13, 2024, from [https://www.who.int/data/gho/data/indicators/indicator-details/GHO/prevalence-of-anaemia-in-children-under-5-years-\(-\)](https://www.who.int/data/gho/data/indicators/indicator-details/GHO/prevalence-of-anaemia-in-children-under-5-years-(-)).

Rebuttal Letter to *Comments Raised by the Reviewers* for the Revised Article
*High-resolution spatial prediction of anemia risk among children between 6 and
59 months in low- and middle-income countries (COMMSMED-24-0035B)*

submitted to

Communications Medicine

January 29, 2025

Comments Reviewer #1

1 Introduction & Summary of the Content

This is a peer-review of the second revision of 'High-resolution spatial prediction of anemia risk among children between 6 and 59 months in low- and middle-income countries' by Seiler, Wetscher, Harttgen, Utzinger, and Umlauf, submitted Nov 2024.

2 Summary of the Content

The authors note a need for high resolution (spatial-temporal) estimates of anemia in children to help monitor progress in anemia reduction and target precision interventions. Further, they note that existing approaches of this specific outcome are limited to individual or small selections of countries. There is good reason to want to model larger swathes of countries concurrently: namely this allows the modeler to leverage the entire, larger dataset to model associations between the covariates and anemia that are shared across all countries.

To do this, they employ Bayesian distributional regression models to estimate the distribution of hemoglobin (Hb) in children of different ages and sexes across a number of low- and middle-income countries and for every year from 2005 to 2021. They use a batchwise backfitting algorithm to select covariates, use flexible distributional models to account for important characteristics (skew) of the Hb distribution that cannot be well-modeled by more often used Normal distributions, and they perform model comparisons to validate the predictive performance of their selected models.

They conclude with a brief discussion of some of the results.

3 Overall Revision Feedback

I again thank the authors for working to address my comments and those from the other reviewers. All my main concerns have been addressed and I would recommend that the paper be accepted.

Response We would like to thank Reviewer #1 (Aaron Osgood-Zimmerman) for his critical but very constructive review of our paper entitled *High-resolution spatial prediction of anemia risk among children between 6 and 59 months in low- and middle-income countries (COMMSMED-24-0035B)*. Please find below our response to his optional minor comment. Once again, we thank him for his valuable feedback.

4 Minor Remarks

I agree that your pixel-level validation, both in- and out-of-sample, suggest a good model fit for your data and suggest that it is capable of reliable predictions in similar out-of-sample settings.

I only have one question for you: do you think it makes sense to show the credible intervals for the pixel estimates (as you've done) or to include the posterior predictive credible intervals?

I can see arguments either way. I think it's clear from the pixel-level validation plot that the 95% of the pixel-level estimates do not contain the DHS reported data observations – but that's to be expected. On the other hand, approximately 95% of the out-of-sample 95% posterior predictive credible intervals should contain the reported data observations. The post. pred. credible intervals will necessarily be wider than the credible intervals for the pixel estimates, and it may make the plots harder to interpret. I would personally be curious to know what percent of your post. pred. credible intervals cover the data, and it may be a useful validation metric for you to check and include, but I leave that to your discretion.

I've enjoyed reading your paper. Thank you for this contribution to precision global health.

Sincerely,

Aaron Osgood-Zimmerman
Assistant Professor of Statistics
Department of Mathematics
Bucknell University

Response We are very grateful to Aaron Osgood-Zimmerman for his critical but generally positive review, which has helped us to improve the manuscript in a number of ways. Our response to his last optional minor comment is below.

Thanks for your thoughtful question. We agree that evaluating the posterior predictive interval (PPI) on out-of-sample data offers valuable insights into model generalizability and predictive performance.

Figure S 15 in the Supplementary Materials illustrates pixel-level validation, showing uncertainty in model estimates based on posterior distributions. These intervals represent credible intervals for point estimates (e.g., hemoglobin concentration per pixel), which are useful for assessing uncertainty at specific spatial locations.

In contrast, the PPI evaluates how well a model predicts new data. As shown in Table 1, approximately 95% of observed out-of-sample realizations fall within the 95% PPI, demonstrating strong predictive performance. In

Table 1: Coverage of the 95% PPI calculated using the out-of-sample data by region.

Region	Percentage outside PPI (%)	Percentage inside PPI (%)
Sub-Saharan Africa	4.7	95.3
South Asia	5.08	94.92
Madagascar	5.18	94.82

Bayesian distributional regression, posterior samples (e.g., 1000 MCMC draws) define the posterior distribution. The 95% PPI, constructed using the 2.5% and 97.5% quantiles, captures both model parameter uncertainty and inherent data variability. Unlike credible intervals, which quantify uncertainty in specific posterior estimates, the PPI provides a broader assessment of predictive accuracy.

Comments Reviewer #3

Remarks to the Author

I would like to thank the authors for their thorough response to my previous comments and their revisions to the manuscript. This has now addressed all of my concerns and I believe has strengthened the manuscript substantially. I think this is an informative and interesting piece of work that adds value to the field. I have no further comments requiring attention.

Response We would like to thank Reviewer #3 for his/her overall positive and constructive feedback, which helped us to substantially improve the paper in several ways. We thank him/her again for his valuable feedback.